# A multi-time scale rolling optimization framework for low-carbon operation of CCHP microgrids with demand response integration

Jue Wang[1,2], Zhiwei Cheng[3], Dejun Lu[4], Mingxiang Zhu[2], Dengfeng Zhang[1]*

1 Institute of Intelligent Manufacturing, Nanjing Tech University, Nanjing, China, 2 Nanjing Normal University Taizhou College, Taizhou, China, 3 Jiangsu Pinggao Tai Shida Electric Co. LTD, Taizhou, China, 4 Special Equipment Safety Supervision Inspection Institute of Jiangsu Province, Taizhou, China

* zhdfnjtech@163.com

## Abstract

Microgrid systems incorporating carbon trading mechanisms and *demand response* (*DR*) demonstrate significant potential for facilitating low-carbon societies and advancing sustainable energy development. The optimal operation of microgrid systems faces challenges due to: (1) response rate disparities among cooling, heating, and power equipment, (2) load prediction inaccuracies, and (3) complex interdependencies in multi-energy device coupling. To address these challenges, we propose a two-layer rolling optimization framework with multi-time scale scheduling for CCHP microgrid systems. First, wind and photovoltaic power generation are predicted using a CNN-ATT-BiLSTM model, with comparative analysis against standalone CNN, BiLSTM and CNN-LSTM models. Second, we establish a multi-time scale optimization model for CCHP-MG systems, with operating cost minimization as the objective function. Finally, we evaluate four operational scenarios incorporating DR and carbon trading mechanisms, with comparative cost analysis. Case study results demonstrate that the proposed model simultaneously satisfies cooling/heating/power demand while mitigating stochastic supply-demand fluctuations through multi-temporal resolution coordination.

## 1 Introduction

Under China's "dual carbon" policy framework, the Ministry of Ecology and Environment (MEE) issued the 2024 Consultation Draft for the National Carbon Emission Trading Scheme, introducing revised allowance allocation mechanisms for the power generation sector. Key revisions include: reduction in carbon emission benchmarks, and shortened compliance periods from biennial to annual cycles. Notably, the scheme introduces quota compensation mechanisms for active peak-shaving participants. These modifications enhance carbon market liquidity while streamlining MRV procedures, consequently reducing verification burdens for enterprises [1–2]. DR

**Data availability statement:** 1. Data Ownership & Access: The dataset used in this study is proprietary and owned by Jiangsu Pinggao Tai Shida Electric Co., LTD. The authors accessed the data under a license agreement specifically for this research. Due to third-party restrictions, we are unable to publicly distribute the raw data. 2. Access for Replication: Interested researchers may request access to the dataset [JIP-24YS (Jiangsu Industrial Park 24-Year Summer Dataset)] by contacting: Mr. BoLiu Email: [272813199@qq.com] Please note that additional terms, fees, or restrictions may apply, as determined by the data provider.

**Funding:** This research was supported by the National Natural Science Foundation of China (Grant No. 62333010 to D.Z.).

**Competing interests:** The authors have declared that no competing interests exist.

programs incentivize user participation in peak load management, simultaneously alleviating grid stress and enhancing energy efficiency. Three key decarbonization mechanisms facilitate optimal resource allocation: grid-integrated renewables, carbon trading markets, and demand response programs [3–5]. CCHP microgrids integrate distributed energy resources (DERs), conventional generators, storage systems, and flexible loads to establish multi-energy supply networks that simultaneously enhance clean energy utilization, satisfy diversified load demands, and mitigate environmental impacts. However, CCHP-MG operational complexity escalates with: renewable intermittency, multi-energy coupling dynamics, and heterogeneous response timescales. Therefore, integrating carbon trading with DR in multi-temporal CCHP-MG optimization is critical for achieving sustainable energy systems and low-carbon transitions [6–10].

Demand response constitutes a grid management paradigm that modulates load profiles during peak periods through price/incentive signals. This response is usually achieved through smart meters, automated control systems, or pre-set tariff mechanisms to balance supply and demand, reduce the peak loads on the grid, improve energy efficiency, and potentially save users on their electricity bills. Ref. [11] conducted research and statistics on the electricity consumption patterns of residential users, industrial and commercial users, and transportation users, etc., analyzed the relationship between the potential DR of different types users and factors such as the electricity consumption patterns, seasons, and the behavior of electricity consumption. Ref. [12] simulated the implementation strategies of different demand response programs and proposed a qualitative analysis method of user DR potential considering the influence of seasonal factors and calling time. The dynamic dispatching process of DR represents a closed-loop control mechanism that enables power systems to dynamically optimize demand-side resources – including flexible loads, energy storage systems, and electric vehicles – in response to real-time grid conditions, pricing signals, or dispatch commands. This sophisticated process achieves multiple objectives: peak shaving, supply-demand balancing, and enhanced system flexibility and economic efficiency. Ref. [13] proposes a tri-level DR framework to enhance the flexibility of electric vehicle charging stations (EVCS) in coupled power-transportation networks (CPTN) by integrating distribution network operators (DNOs). The framework employs deep reinforcement learning (DRL) to dynamically balance profit maximization and operational safety, the results demonstrate superior EVCS operational flexibility compared to traditional charging load regulation methods. Ref. [14] integrates micro-urban transportation network (MUTN) with distribution network (PDN) to realize high-precision vehicle-to-grid (V2G) analysis. It combines SUMO for MUTN simulation and optimized DistFlow model for PDN analysis, and combines the detailed charging dynamics of fast charging station (FCS) and slow charging station (SCS). Ref. [15] proposes a resilient hierarchical load frequency control (LFC) strategy for microgrids (MGs) integrating distributed renewable energy sources (DRESs), particularly addressing wind power fluctuations. Ref. [16] proposes a novel distributed hybrid-triggered (HTed) secondary control (SC) scheme for DC microgrids (MGs) that combines event-triggered communication with self-triggered sampling/computation to

optimize efficiency. According to the load response characteristics, DR is divided into price demand response (PDR) and incentive demand response (IDR), as shown in Table 1. PDR regulates electricity consumption plans by changing electricity prices to guide users to make reasonable electricity consumption behaviors. IDR encompasses contractual arrangements including: interruptible loads, demand bidding, emergency response protocols, and direct load control systems.

A high proportion of renewable energy sources (RES) is the most important measure for the power industry to achieve energy efficiency and "dual carbon" targets, while wind power and photovoltaic power are vulnerable to a variety of natural factors, highly intermittent and uncertain. Therefore, large-scale wind power and photovoltaic grid integration can not only threaten the reliability and stability of the power grid system, but also bring more unpredictable uncertainties to power dispatching. At present, various wind power forecasts can be categorized according to their time scales [17]. According to the different prediction principles, it can be divided into physical methods, statistical methods, machine learning methods and hybrid methods [18–19]. With the development of AI, many machine learning-based methods have also been used for wind power prediction, such as convolutional neural networks (CNN), Deep Belief Networks (DBN), Auto Encoder (AE), Recurrent Neural Networks (RNN), Gated Recurrent Unit (GRU), and long short-term memory (LSTM). CNN can efficiently extract nonlinear local features of wind power data, and LSTM is often used in the field of wind power prediction because of its special memory ability to capture and utilize long-term and short-term time dependencies. Ref. [20] used LSTM for modeling, which prediction error was greatly reduced and the prediction accuracy was improved on a long-term scale. Bidirectional long short-term memory (BiLSTM) was also proposed, which can capture better bidirectional dependencies. BiLSTM can recognize time series features present in data from the past to the future on the one hand, and inverse time series features from the future and the past on the other hand [21–24].

CCHP-MG multi-time scale optimal scheduling refers to a comprehensive strategy for energy distribution and management of renewable energy, generator devices, energy storage equipment, other energy equipment and various loads. It takes everything from real-time operations to intraday and day-to-day planning into account, and involves dynamically adjusting the balance of power supply and demand to maximize efficiency, reduce costs, and guarantee the quality and reliability of power supply. Ref. [25] proposed a probabilistic energy management approach considering both day-ahead scheduling and real-time scheduling. Ref. [26] established a multi-temporal and spatial scale optimization model for IES containing multiple communities, which takes advantage of the complementary strengths of multiple energy sources and promotes the balance between energy supply and demand. Ref. [27] established a three-stage optimization model of day-ahead-intra-day rolling-real-time adjustment, and took the lowest daily operating cost, the lowest cost of energy purchase and penalty cost of unit output change in the rolling time, and the smallest total adjustment rate of equipment power as objectives, so as to obtain the real-time smoothed output of the equipment. The real-time smoothing power plan was obtained. Ref. [28] proposed a two-tier coordinated energy management approach, which was applied to smart multi-microgrids to enable flexible and secure operation of microgrids. Ref. [29] considered the time scale of matching the temporal characteristics with the scheduling peaks, and proposed an adaptive scheduling strategy being applied to the

**Table 1. Class of electric DR.**

| Class | | Response mechanism | Response time |
|---|---|---|---|
| PDR | | PDR guides users to use electricity reasonably by changing the price of electricity. The amount of load change in the day-ahead scheduling stage. | Day-ahead (1h) |
| IDR | Class A | The response time of this kind of load is > 1h, and users need to be informed one day in advance, and the call plan should be determined in the dispatching day-ahead scheduling stage. | Day-ahead (1h) |
| | Class B | The response time of this kind of load is 5–15 min, and users need to be informed 15 min-4h in advance. The call plan in the intra-day scheduling stage. | Intra-day (15 min) |
| | Class C | This kind of load can respond in real-time, and users need to be informed 5–15 minutes in advance, and the calling plan should be determined in the real-time scheduling stage. | Real-time (5 min) |

scheduling cycle of an integrated energy system. There is a lack of research on two-layer models at multi-time scales, based on differences in response rates of cooling, heating, and power-related equipment.

The operational complexity of CCHP-MG systems escalates due to three key factors: high renewable penetration, load demand uncertainty, and heterogeneous response timescales across energy subsystems. To address these challenges, we develop a multi-temporal CCHP-MG optimization framework with the following innovations:

(1) A comprehensive equipment modeling framework minimizes operational costs while incorporating an attention-enhanced CNN-BiLSTM hybrid model for renewable forecasting, outperforming baseline CNN and BiLSTM architectures.

(2) A bilevel optimization architecture accounts for heterogeneous response dynamics: thermal energy dispatch operates at 15-min intervals (upper layer) while electrical power adjusts at 5-min resolutions (lower layer), enabling prediction error compensation, day-ahead plan recalibration, and system stabilization.

(3) Real-time scheduling (1-min resolution) coordinates fuel cells, energy storage, and grid interactions to mitigate stochastic fluctuations through model predictive control.

(4) Case studies demonstrate cost reductions across four scenarios integrating DR and carbon trading, while the multi-time-scale approach simultaneously ensures 100% energy demand satisfaction and fluctuation attenuation in supply-demand mismatches.

## 2 The structure of CCHP-MG

In this paper, the CCHP-MG system considers four energy forms: cold, heat, power, and gas, including energy supply equipment, energy storage equipment, and energy conversion equipment. The framework is shown in Fig 1. The energy equipment in the system mainly includes photovoltaic power (PV), wind turbine (WT), fuel cell (FC), micro-gas turbine

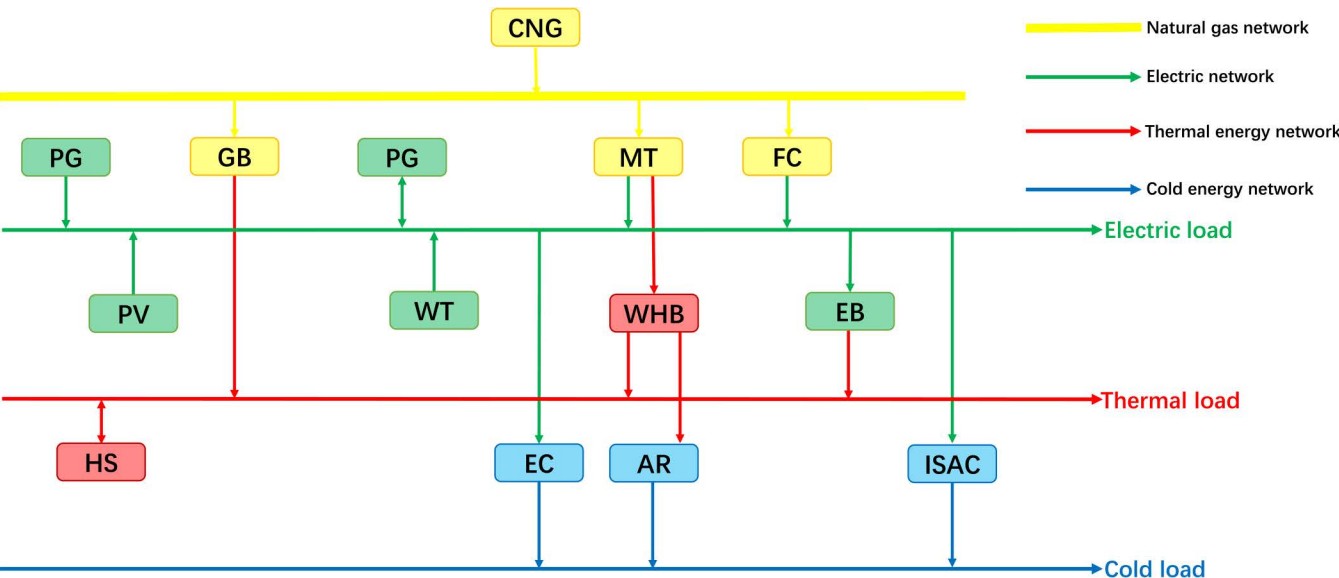

**Fig 1 The Structure of CCHP-MG**

**Fig 1.  The structure of CCHP-MG.**

(MT) and gas boiler (GB); Energy storage equipment mainly includes energy storage (ES), cold storage (CS), and heat storage (HS); Energy conversion equipment includes waste heat boiler (WHB), electric boiler (EB), ice-storage air-conditioning (ISAC), absorption refrigerator (AR) and so on [30–31].

## 2.1 MT

When MT is in operation, the exhaust high-temperature waste heat flue gas is recovered by WHB and then heated and cooled by HE and AR. The characteristic model of MT is roughly depicted in the following:

$$G_{MT}^T = \frac{P_{MT}^T(1 - \eta_{MT} - \eta_L)}{\eta_{MT}} \tag{1}$$

$$H_{MT}^T = G_{MT}^T C_{OP,h} \eta_h \tag{2}$$

$$Q_{MT}^T = G_{MT}^T C_{OP,c} \eta_c \tag{3}$$

where $G_{MT}^T$ is the exhaust residual heat of MT in the T period; $P_{MT}^T$ is the output power of MT in the T period; $\eta_{MT}$ is the power generation efficiency of MT; $\eta_L$ is the heat dissipation loss coefficient; $H_{MT}^T$ and $Q_{MT}^T$ are heating and cooling capacity of HE and AR in the T period, respectively; $C_{OP,h}$ and $C_{OP,c}$ are heating coefficient and cooling coefficient, respectively; $\eta_h$ and $\eta_c$ are flue gas recovery rates for heating and cooling, respectively.

At the same time, MT has to satisfy the upper and lower bound constraints and climb rate constraints:

$$U_{MT}^T P_{MT}^{\min} \leq P_{MT}^T \leq U_{MT}^T P_{MT}^{\max} \tag{4}$$

$$P_{MT}^{down} \leq P_{MT}^T - P_{MT}^{T-1} \leq P_{MT}^{up} \tag{5}$$

where $U_{MT}^T$ is the start-stop state marker of the MT, and its value of 0 indicates the shutdown, A value of 1 indicates that the device is turned on. $P_{MT}^{\min}$ and $P_{MT}^{\max}$ are the minimum and maximum output power of MT, respectively; $P_{MT}^{down}$ and $P_{MT}^{up}$ are the upper and lower limits of the climb rate of MT, respectively.

The formula for the fuel consumption $C_{MT}^T$ of MT, where $C_{GAS}$ is the price of natural gas, $Q_H$ is the calorific value, and $\beta_{MT}$ is the fuel coefficient for starting and stopping MT.

$$C_{MT}^T = C_{GAS} \left( \frac{P_{MT}^T}{\eta_{MT} Q_H} + \beta_{MT} U_{MT}^T \right) \tag{6}$$

## 2.2 FC

Fuel cells convert chemical energy stored in hydrogen-containing fuels such as natural gas, methanol, and oxygen into electricity efficiently and pollution-free. In this paper, a proton exchange membrane fuel cell (PEMFC) is used, which uses natural gas as the primary energy source. Since the fuel cell is used electrical dispatching micropower source, the rest of the heat utilization is not considered.

$$P_{FC}^T = F_{FC}^T \eta_{FC} \tag{7}$$

$$P_{FC}^{\min} \leq P_{FC}^T \leq P_{FC}^{\max} \tag{8}$$

where $F_{FC}^T$ is the consumption of natural gas, $\eta_{FC}$ is the efficiency of FC, $P_{FC}^T$ is the output power of FC in the T period, $P_{FC}^{\min}$ and $P_{FC}^{\max}$ is the minimum and maximum output, respectively. $C_{FC}^T$ is the fuel cost of the fuel cell in the period T.

$$C_{FC}^T = C_{GAS} \frac{P_{FC}^T}{\eta_{FC} Q_H} \tag{9}$$

## 2.3 ES

In this paper, the energy storage device (ES) uses a battery to store electrical energy. The battery should not only constrain the amount of charging and discharging power, but also consider avoiding low power and low state of charge (SOC), as in Eq. (10). The battery also meets the charge and discharge constraints in its operation, as shown in Eq. (11).

$$\begin{cases} S_{SOC}^T = S_{SOC}^{T-1} + \left( \eta_{bt,chr} P_{bt,chr}^T - \frac{P_{bt,dis}^T}{\eta_{bt,dis}} \right) \Delta T \\ S_{SOC}^{\min} \leq S_{SOC}^T \leq S_{SOC}^{\max} \end{cases} \tag{10}$$

$$\begin{cases} U_{bt,chr}^T P_{bt,chr}^{\min} \leq P_{bt,chr}^T \leq U_{bt,chr}^T P_{bt,chr}^{\max} \\ U_{bt,dis}^T P_{bt,dis}^{\min} \leq P_{bt,dis}^T \leq U_{bt,dis}^T P_{bt,dis}^{\max} \end{cases} \tag{11}$$

where: $S_{SOC}^T$ is the SOC value of the ES; $P_{bt,chr}^T$ and $P_{bt,dis}^T$ are the charging and discharging power, respectively; $\eta_{bt,chr}$ and $\eta_{bt,dis}$ are the charging and discharging efficiency, respectively; $\Delta T$ is the time interval; $U_{bt,chr}^T$ and $U_{bt,dis}^T$ are the charging and discharging state markers, whose values are 0 for shutdown and 1 for operation, and satisfy the mutual exclusion constraint.

$$U_{bt,dis}^T + U_{bt,chr}^T \leq 1 \tag{12}$$

In addition, the climb rate constraints of charge and discharge described in Eq. (13) must be satisfied, where $P_{bt,chr}^{up}$, $P_{bt,chr}^{down}$ and $P_{bt,dis}^{up}$, $P_{bt,dis}^{down}$ are the upper and lower limits of the climb rate in the charging and discharging states, respectively.

$$\begin{cases} P_{bt,chr}^{down} \leq P_{bt,chr}^T - P_{bt,chr}^{T-1} \leq P_{bt,chr}^{up} \\ P_{bt,dis}^{down} \leq P_{bt,dis}^T - P_{bt,dis}^{T-1} \leq P_{bt,dis}^{up} \end{cases} \tag{13}$$

## 2.4 EB

EB can convert power energy into heat energy to meet the user's heat load demand, and its constraints are shown in the following:

$$H_{EB}^T = P_{EB}^T \eta_{EB} \tag{14}$$

$$0 \leq P_{EB}^T \leq P_{EB}^{\max} \tag{15}$$

where $P_{EB}^T$ and $H_{EB}^T$ are the power energy and heat energy of EB in the T period, respectively; $P_{EB}^{\max}$ is the maximum permissible capacity of EB; $\eta_{EB}$ is the efficiency coefficient of EB.

## 2.5 GB

The gas boiler supplements the heat energy when the heat supply is insufficient, and its model is given in Eqs. (16), (17) and (18). The output heat energy can be expressed as $H_{GB}^T$. $F_{GB}^T$ is the consumption of natural gas, $\eta_{GB}$ is the efficiency of GB. $H_{GB}^{\min}$ and $H_{GB}^{\max}$ are the minimum and maximum thermal energy output, respectively.

$$H_{GB}^T = F_{GB}^T \eta_{GB} \tag{16}$$

$$H_{GB}^{min} \leq H_{GB}^T \leq H_{GB}^{max} \tag{17}$$

$$C_{GB}^T = C_{GAS} F_{GB}^T / Q_H \tag{18}$$

## 2.6 HS

HS uses heat storage tank for heat energy storage, and the constraints are as follows:

$$\begin{cases} S_{tst}^T = S_{tst}^{T-1}(1-\gamma_h) + \left(\eta_{tst,chr} H_{tst,chr}^T - \frac{H_{tst,dis}^T}{\eta_{tst,dis}}\right) \\ S_{tst}^{min} \leq S_{tst}^T \leq S_{tst}^{max} \end{cases} \tag{19}$$

$$\begin{cases} U_{tst,chr}^T H_{tst,chr}^{min} \leq H_{tst,chr}^T \leq H_{tst,chr}^{max} U_{tst,chr}^T \\ U_{tst,dis}^T H_{tst,dis}^{min} \leq H_{tst,dis}^T \leq H_{tst,dis}^{max} U_{tst,dis}^T \end{cases} \tag{20}$$

where $S_{tst}^T$ is the stored heat energy of HS; $H_{tst,chr}^T$ and $H_{tst,dis}^T$ are charging and discharging heat energy of HS, respectively; $\gamma_h$ is the energy self-loss rate of HS. $\eta_{tst,chr}$ and $\eta_{tst,dis}$ are efficiencies of charging and discharging heat energy, respectively; $U_{tst,chr}^T$ and $U_{tst,dis}^T$ are the charging and discharging heat energy state markers of HS, respectively.

$$U_{tst,dis}^T + U_{tst,chr}^T \leq 1 \tag{21}$$

$$\begin{cases} H_{tst,chr}^{down} \leq H_{tst,chr}^T - H_{tst,chr}^{T-1} \leq H_{tst,chr}^{up} \\ H_{tst,dis}^{down} \leq H_{tst,dis}^T - H_{tst,dis}^{T-1} \leq H_{tst,dis}^{up} \end{cases} \tag{22}$$

## 2.7 ISAC

The CS uses a cold storage tank for cold storage, and the ISAC stores cold when it is running in the ice state. The following constraints need to be met.

$$U_a^T Q_a^{min} \leq Q_a^T \leq Q_a^{max} U_a^T \tag{23}$$

$$0 \leq Q_c^T \leq Q_a^{max} U_c^T \tag{24}$$

$$Q_a^{min} \leq Q_c^T + Q_a^T \leq Q_a^{max} \tag{25}$$

$$0 \leq Q_d^T \leq Q_d^{max} U_d^T \tag{26}$$

$$S_{ice}^T = S_{ice}^{T-1}(1-\gamma_Q) + \left(\eta_{ice,chr} Q_c^T - \frac{Q_d^T}{\eta_{ice,dis}}\right) \tag{27}$$

$$S_{ice}^{down} \leq S_{ice}^T - S_{ice}^{T-1} \leq S_{ice}^{up} \tag{28}$$

In the above equations, $Q_a^T$, $Q_c^T$ and $Q_d^T$ are the output cooling energy, ice storage energy and ice melting energy of ISAC, respectively; $U_a^T$, $U_c^T$ and $U_d^T$ are cooling state, ice storage state and ice melting stage markers of ISAC. $S_{ice}^T$ is the cooling

energy stored in CS; $\gamma_Q$ is the self-loss coefficient; The ice storage coefficient and ice melting coefficient are $\eta_{ice,chr}$ and $\eta_{ice,dis}$, respectively; $S_{ice}^{up}$ and $S_{ice}^{down}$ are the upper and lower limits of the climb rate of SC, respectively.

## 3  WT and PV power prediction

### 3.1  WT

The ideal formula for the output of WT is shown in Eq. (29):

$$P = \frac{1}{2}\eta_P \rho A V^3 \tag{29}$$

$\eta_P$ is the wind energy utilization efficiency of WT, and $\rho$ the air density; $A$ is the swept area; $V$ is the wind speed; $\rho$ is determined by temperature, air pressure, and humidity together, which can be expressed by:

$$\rho = 3.48\frac{p}{T}\left(1-0.378\frac{\Phi p_b}{p}\right) \tag{30}$$

$p$ is the standard atmospheric pressure, $p_b$ is the saturated water vapor pressure; T is the thermodynamic temperature; $\Phi$ for the relative humidity of the air; Therefore, when considering the influence of $\eta_P$ on the output, air pressure, temperature and humidity are actually also considered. Finally, in addition to the indirect factors that affect the output power of wind power discussed in the above equation, wind direction is also a particularly important factor affecting the output.

### 3.2  PV

The output power of PV can be expressed as:

$$P_{PV} = \eta_{PV}SI[1-0.005(T_0 + 25)] \tag{31}$$

where $\eta_{PV}$ represents the conversion efficiency of solar panels; $S$ represents the area of the PV cell, $I$ represents the intensity of solar radiation, and $T_0$ represents the operating temperature of the PV module. In addition, PV power generation will also be affected by other factors, such as solar irradiance, temperature, humidity, weather and season.

### 3.3  Predictive model

WT and PV power generation have non-linear, random and other characteristics, many influencing factors will lead to changes. In summary, the WT power prediction is analyzed by selecting wind speed, wind direction, air pressure, temperature, humidity and historical power data as the input data set. The PV power prediction data quantitatively analyze the correlation between PV power and meteorological factors based on Pearson's correlation coefficient, and the Global Horizontal Irradiance (GHI), zenith angle, temperature, relative humidity, cloud opacity, wind speed, and the historical power data are selected to carry out PV power prediction studies. The selected data is normalized and fed into a predictive model [32–33].

In this paper, a two-layer CNN network is used to mine and extract the potential connection of data, which includes two layers of two-dimensional convolutional layer and two layers of pooling layer alternate composition. A dropout layer is added to prevent overfitting. Then, the important feature information is extracted through the Attention module, and then the temporal features of the data are extracted by two-layer BiLSTM. Finally, the final predicted power is inputted to the fully connected layer by the Attention module, which completes the construction of the CNN-ATT-BiLSTM model. The framework of the model is shown in Fig 2.

CNN and BiLSTM are selected to compare with the proposed model in this paper, and the predicted waveforms of WT and PV power are shown in Figs 3 and 4.

The proposed CNN-ATT-BiLSTM model can better track the variation of WT and PV power. The prediction errors by three models are shown in Table 2:

The comparative analysis of prediction performance, as illustrated in 3 and 4 Figs and Table 2, demonstrates that CNN-LSTM achieves superior performance to the standalone CNN and BiLSTM model, exhibiting reduced prediction error and enhanced accuracy. However, the proposed CNN-ATT-BiLSTM architecture emerges as the most effective model, with the minimal prediction error and an $R^2$ value approaching unity, indicating optimal predictive capability.

This advanced hybrid model integrates three complementary mechanisms: CNN for extracting local spatial features, BiLSTM networks for capturing comprehensive temporal dependencies in both forward and backward directions, and ATT for dynamically emphasizing critical temporal features. The CNN-LSTM architecture, while maintaining spatial feature extraction capabilities through its CNN component and simplified structure, demonstrates effective performance for stationary time series but fails to account for reverse temporal dependencies. The standalone CNN model shows proficiency in identifying local patterns and periodic fluctuations, yet exhibits limitations in modeling long-term temporal relationships. The BiLSTM network, although capable of bidirectional temporal context learning, lacks spatial feature extraction capacity.

These comparative results clearly demonstrate the superior modeling capability of the CNN-ATT-BiLSTM architecture, which effectively combines the strengths of spatial feature extraction, bidirectional temporal modeling, and attention-based feature weighting. The model's exceptional performance is particularly evident in renewable energy power prediction applications, where it successfully captures both spatial correlations and complex temporal dynamics simultaneously.

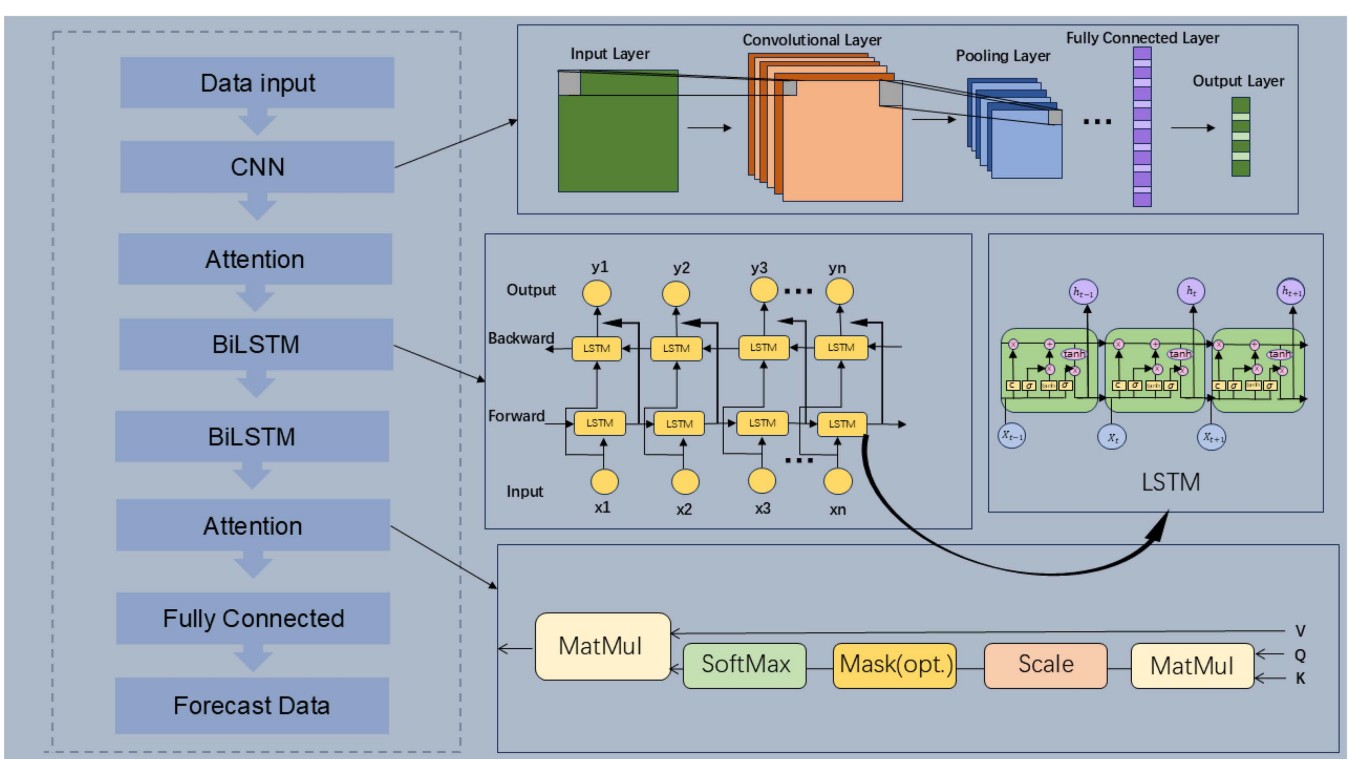

**Fig 2. The Framework of Predictive Model.**

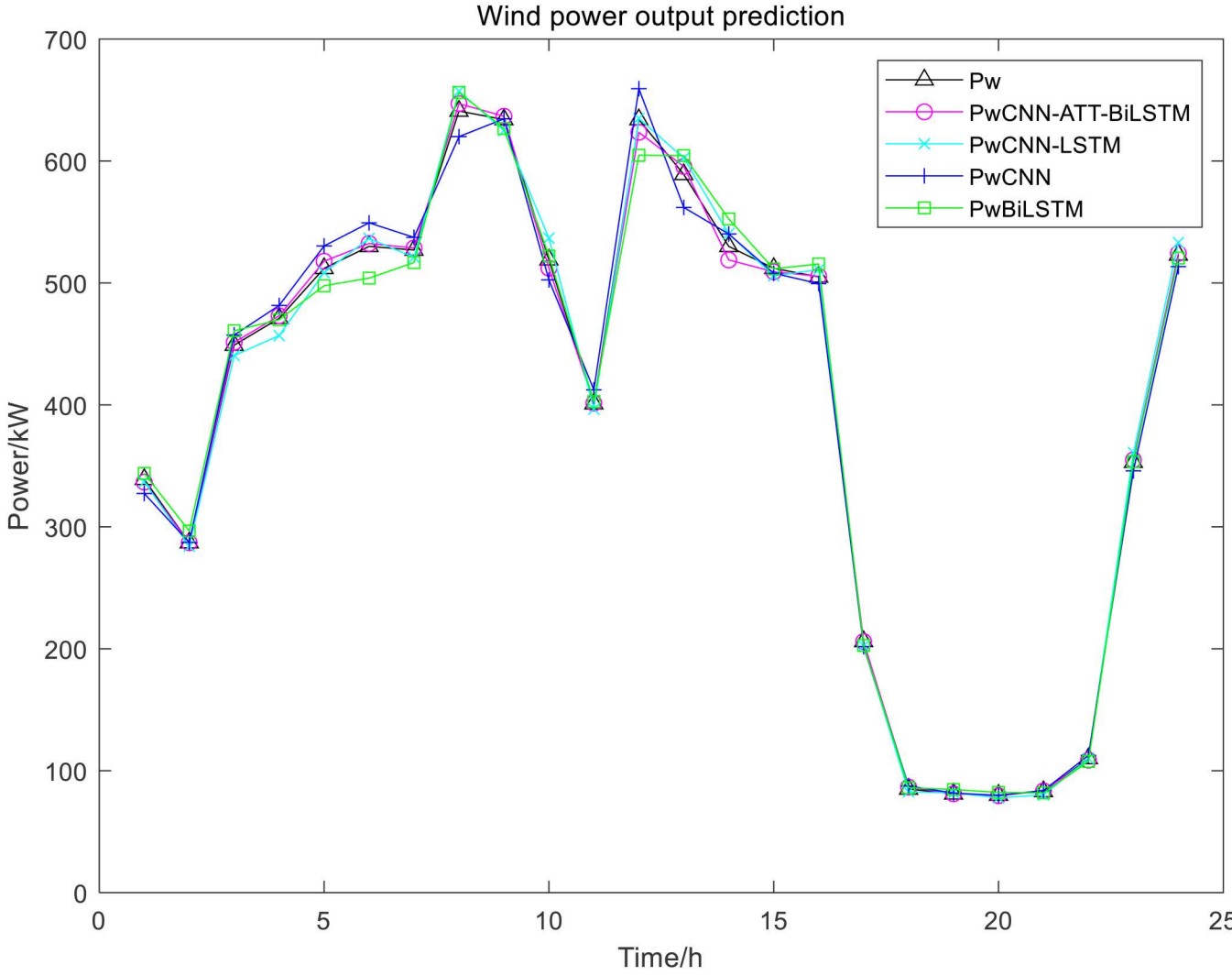

**Fig 3. The Predicted Waveforms of WT Power.**

## 4 Multi-time scale scheduling operation

### 4.1 Multi-time scale Model Predictive Control

To address the multi-time scale scheduling challenges in IES, numerous forecasting methods and optimization techniques – extending beyond Model Predictive Control (MPC) – can be utilized to improve scheduling performance. Table 3 presents a comparative analysis of prevalent forecasting approaches along with their distinctive application characteristics. Based on the comparative evaluation of prediction model attributes presented in the table, MPC has been selected as the most suitable approach for this study.

MPC is an advanced online control algorithm that optimizes system performance by predicting future outputs and states within a finite time horizon. This algorithm generates an optimal control sequence by continuously solving a constrained optimization problem based on real-time system states and environmental variations, thereby demonstrating

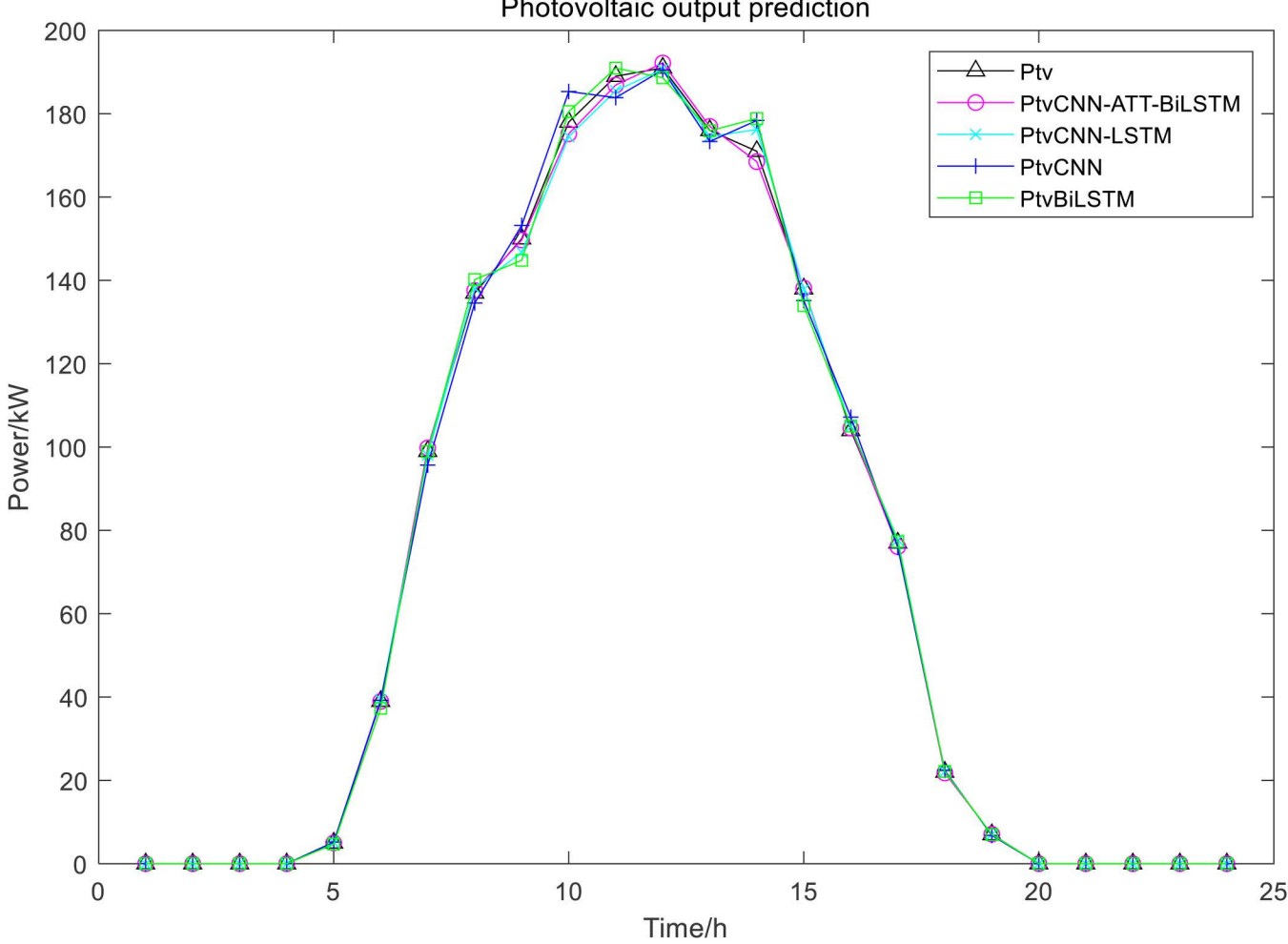

**Fig 4. The Predicted Waveforms of PV Power.**

**Table 2. Prediction errors of different models.**

|  | Prediction model | RMSE(kW) | MAE(kW) | R²/% |
|---|---|---|---|---|
| WT power | CNN | 24.314 | 16.538 | 0.892 |
|  | BiLSTM | 18.424 | 11.436 | 0.924 |
|  | CNN-LSTM | 13.825 | 7.29 | 0.968 |
|  | CNN-ATT-BiLSTM | 11.937 | 6.127 | 0.981 |
| PV power | CNN | 12.314 | 7.538 | 0.902 |
|  | BiLSTM | 9.424 | 6.936 | 0.931 |
|  | CNN-LSTM | 7.902 | 5.834 | 0.976 |
|  | CNN-ATT-BiLSTM | 7.137 | 5.127 | 0.987 |

**Table 3. Comparison of Prediction Methods in Multi-Timescale Scheduling.**

| Comparison of Application Scenarios | | | |
|---|---|---|---|
| Forecasting methodology | Applicable time scale | Advantages | Limitations |
| MPC (Model Predictive Control) | Real-time/ Intra-day | Rolling optimization, strong constraint handling | High computational complexity |
| LSTM (Long Short-Term Memory) | Day-ahead | Captures long-term dependencies | Requires large training datasets |
| ARIMA (Autoregressive Integrated Moving Average) | Short-Term/ Medium-Term | Simple, highly interpretable | Poor adaptability to nonlinear data |
| DRL (Deep Reinforcement Learning) | Joint Prediction-Decision Optimization | Enables end-to-end optimization | Unstable training, hyperparameter sensitivity |

superior control performance. For a multi-device system comprising N components, the predicted output sequence group at time step k for future j time steps can be mathematically expressed as follows:

$$y_i(k) = \left[ y_i\left(k+1\,\middle|\,k\right), y_i\left(k+2\,\middle|\,k\right), \ldots, y_i\left(k+j\,\middle|\,k\right) \right] \tag{32}$$

The sequence of corresponding control variables is as follow:

$$\Delta u_i(k) = [u_i(k+1) - u_i(k), \ldots, u_i(k+j) - u_i(k+j-1)] = [\Delta u_i(k), \ldots, \Delta u_i(k+j-1)] \tag{33}$$

The system state variable sequence can be expressed as $x_i(k) = [y_i(k), u_i(k)]$. The system constraints to be satisfied are as follows:

$$\begin{cases} \Delta u_{i,\min}(k+j) \leq \Delta u_i(k+j) \leq \Delta u_{i,\max}(k+j) \\ \Delta u_{i,\min}(k+j) \leq \Delta u_i(k+j+1) - \Delta u_i(k+j) \leq \Delta u_{i,\max}(k+j) \\ y_{i,\min}(k+j) \leq y_i(k+j) \leq y_{i,\max}(k+j) \end{cases} \tag{34}$$

Where: $u_i(k+j)$ and $y_i(k+j)$ are the control and output variables of the i-th device at time $k+j$, respectively; $u_{i,\max}(k+j)$, $u_{i,\min}(k+j)$, and $y_{i,\max}(k+j)$, $y_{i,\min}(k+j)$ are the upper and lower constraints on the control and output variables at time $k+j$, respectively; $\Delta u_{i,\max}(k+j)$ and $\Delta u_{i,\min}(k+j)$ are the upper and lower constraints on the incremental amount of the control at time $k+j$, respectively.

The overall control strategy of the system can be expressed as follows:

$$F = \min \sum_{m=1}^{j} P \left( y\left(k+m\,\middle|\,k\right) - y_r\left(k+m\,\middle|\,k\right) \right)^2 + Q \left( u(k+m) - u(k+m-1) \right)^2 \tag{35}$$

Where: $y_r\left(k+m\,\middle|\,k\right)$ is the reference trajectory of the system at $k+m$ moments; $y\left(k+m\,\middle|\,k\right)$ and $u(k+m-1)$ are the predicted output variables and control variables of the system at $k+m$ time, respectively; P and Q are the weight coefficient matrices of the output variables and control variables, respectively. In this study, the system outputs are defined as the electrical power purchased from the grid and the natural gas consumption. The control variables consist of the power outputs from MT, FC, ES, EB, GB and IASC. The predicted load variations, along with the forecasted power generation fluctuations from wind turbines and photovoltaic systems, are treated as system disturbances. Based on these definitions, a multi-input multi-output (MIMO) state-space model is established for the energy management system.

The multi-time dispatching model proposed mainly includes day-ahead dispatching, intra-day rolling and real-time scheduling. Its multi-time scale optimal dispatching model is shown in Fig 5.

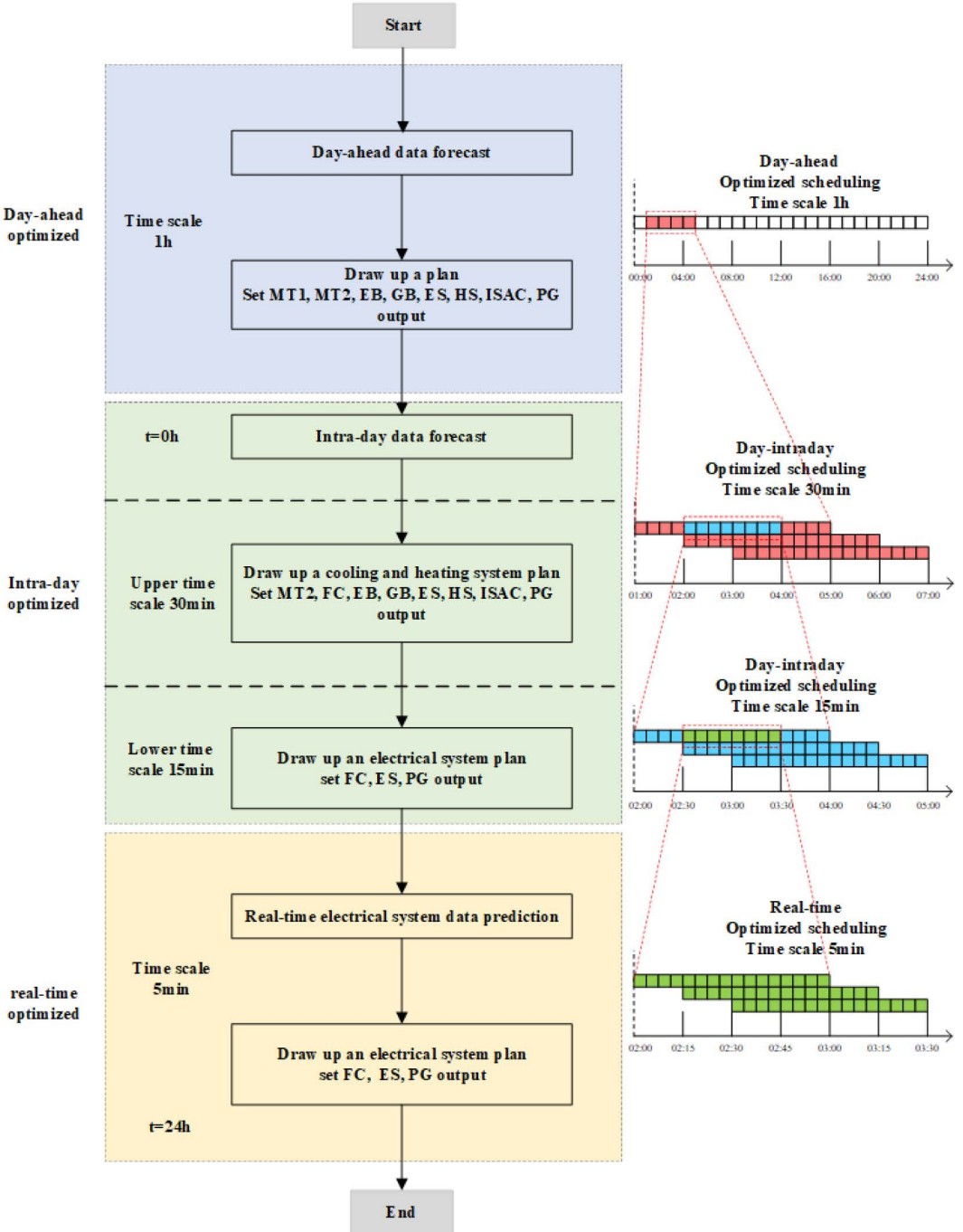

**Fig 5. Multi-time scale optimal dispatching model.**

The day-ahead plan is formulated 24 h ahead of time, and is 1 h time scale. In the day-ahead dispatching, determine the start-stop and output of MT1, MT2, FC, GB, EB, ES, HS, ISAC, PG, PDR and class A IDR load invocation, and substitute it into the intra-day dispatching.

The intra-day rolling optimization considers the correlation and complementarity of cooling, heating and power, and dispatches the cooling and heating loads on the upper-layer, so that the electrical load is scheduled on the lower-layer. The cooling and heating load is predicted and scheduled every 30 min, and the prediction time range is the data of the next 4 h in the scheduling period. The outputs of $MT_2$, GB, EB, HS and ISAC are adjusted to meet the cooling and heating load demand of intra-day. The intra-day electric energy is selected and forecasted every 15 min, and the forecast range is the power data of the next 2 h. The outputs of FC, ES and PG is adjusted to meet the class B IDR load invocation plan and the fluctuation of electric load.

The real-time rolling optimization scheduling is selected to forecast and dispatch once in 5 min, and the forecast range is the power data in the next 1 h. The purpose is to use the intraday dispatch plan as a benchmark to fine-tune the FC, ES, PG output and call the class C IDR to correct the intra-day dispatch plan and real-time working conditions [34–35].

This study presents a scenario-based MPC framework to address uncertainty in renewable energy systems while ensuring alignment between Real-time operational, Intra-day and Day-ahead optimization objectives. A double-layer forecasting approach is adopted for intra-day predictions, effectively mitigating the cost inefficiencies caused by myopic decision-making. Additionally, energy storage systems are leveraged to further compensate for forecasting errors and enhance operational flexibility. In the intra-day adjustment layer, MT2 is designated as the slack variable to absorb power imbalances, whereas in the real-time control layer, FC serves as the slack variable to ensure instantaneous power balance.

## 4.2 DR and Carbon trading

DR resources can change the load curve by optimizing the output of devices with different carbon intensity to achieve a reduction in carbon emissions [36]. PDR can transfer some peak load to off-peak time region, thus effectively reduce the carbon emissions of CCHP-MG. In this paper, the PDR uses the time-of-use electricity price in the peak-valley and flat periods, and uses the impact of the electricity price change rate on the load response rate is described by elastic matrix.

$$\begin{bmatrix} \lambda q_1 \\ \lambda q_2 \\ \lambda q_3 \end{bmatrix} = E \begin{bmatrix} dp_1 \\ dp_2 \\ dp_3 \end{bmatrix}$$

(36)

where $E$ is the price demand elasticity matrix, parameter is shown in Table 4; $\lambda q$ is the load rate change matrix; $dp$ is the matrix of the rate of change electricity price. Since the adjustment amount of the user's electricity plan is limited, we set the PDR response amount not exceeding 15% of the load. Fig 6 shows the day-ahead electrical load waveform and the electrical load waveform considering PDR. It is evident that the power load waveform considering the PDR has a clear trend of peak shaving and valley filling.

At the same time, by IDR with different response speeds during the day-ahead and intra-day, load demand can be increased during off-peak hours to absorb WT and PV power, optimizing operational costs. The flexibility of system scheduling and processing problems can be improved by Calling IDR in the real-time stage, such as abandoned WT and PV power caused by WT power, PV power and load forecasting errors. The IDR load calls in this paper are subject to the following limitations:

$$\begin{cases} 0 \leq |\Delta P_{AIDRT}| \leq P_{AIDR_{max}} \\ |\Delta P_{AIDR_T} - \Delta P_{AIDR_{T-1}}| \leq R_{IDRA} \end{cases}$$

(37)

**Table 4. Price demand elasticity matrix.**

| Time frame | Peak | Shoulder | Valley |
|---|---|---|---|
| Peaks | 0.1 | 0.016 | 0.012 |
| Shoulder | 0.016 | 0.1 | 0.01 |
| Valley | 0.012 | 0.01 | 0.1 |

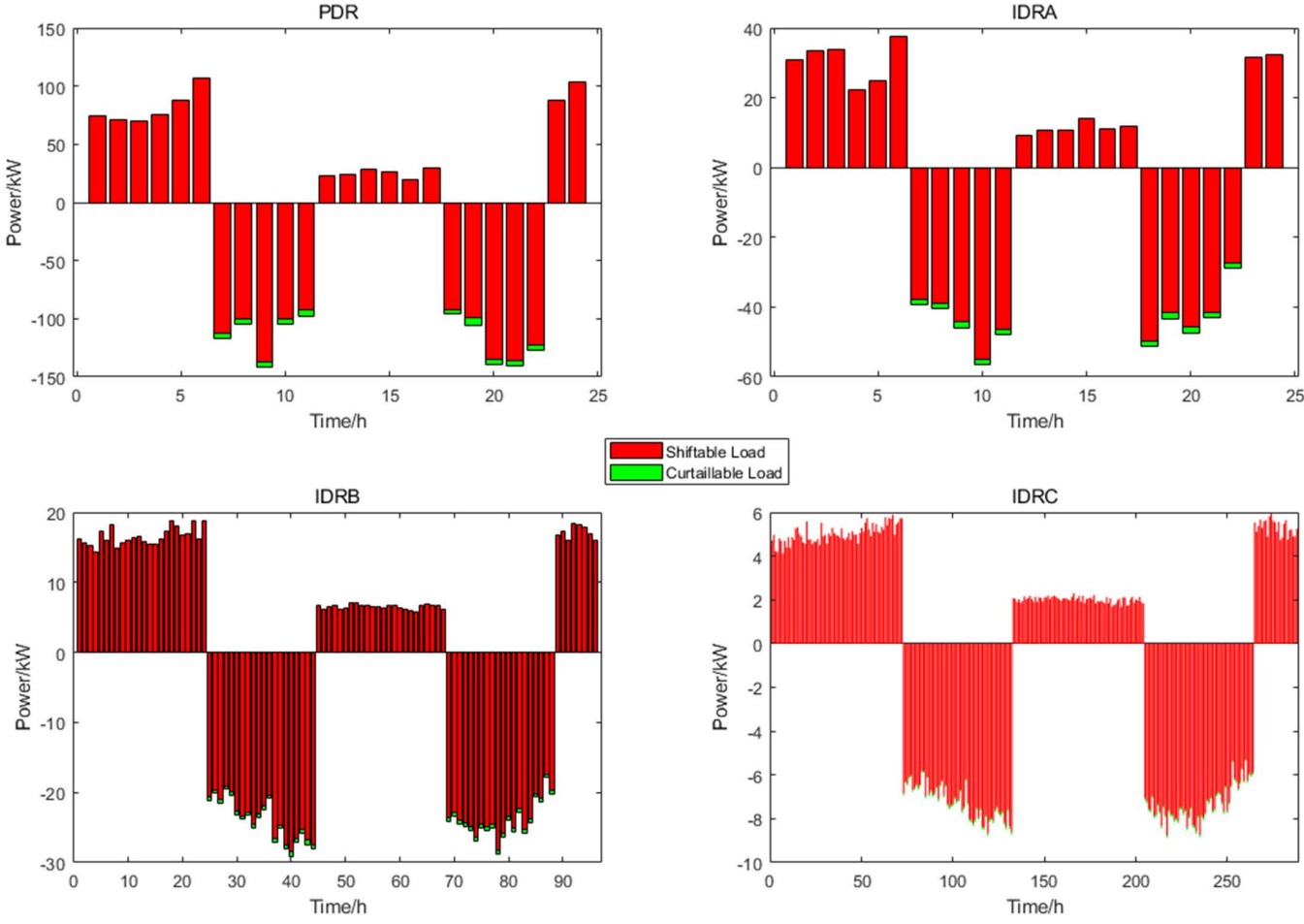

**Fig 6. PDR and IDR Power at different time scales.**

where $P_{AIDR_{max}}$ represents the maximum response amount of class A IDR load. $R_{IDRA}$ is the response speed. This paper sets the capacity of class A, B, and C IDR not exceeding 10%, 5%, and 3% of the total load, respectively. $C_{AIDR}^T$ is the cost. $K_{AIDR}$ is the cost coefficient. $P_{AIDR}$ is the call volume. The cost formula for class B and class C IDR load is similar to that of class A IDR.

$$C_{AIDR}^T = K_{AIDR}P_{AIDR} \tag{38}$$

Fig 6 presents the power dispatch diagrams for day-ahead PDR, day-ahead IDRA, intra-day IDRB, and real-time IDRC. The red segments indicate shiftable loads during the current period, while the green segments represent curtailable loads. Owing to fluctuations in time-of-use electricity pricing, load demand, and renewable energy generation, the demand-responsive loads consistently shift from peak-price periods to valley-price and shoulder-price periods. Fig 7 compares the PDR power profile with the aggregated power profiles of the three IDR types. From these results, the baseline power load profile – both with and without demand response – can be derived, as illustrated in Fig 8. This approach effectively achieves peak shaving and valley filling while facilitating renewable energy integration and stabilizing power fluctuations.

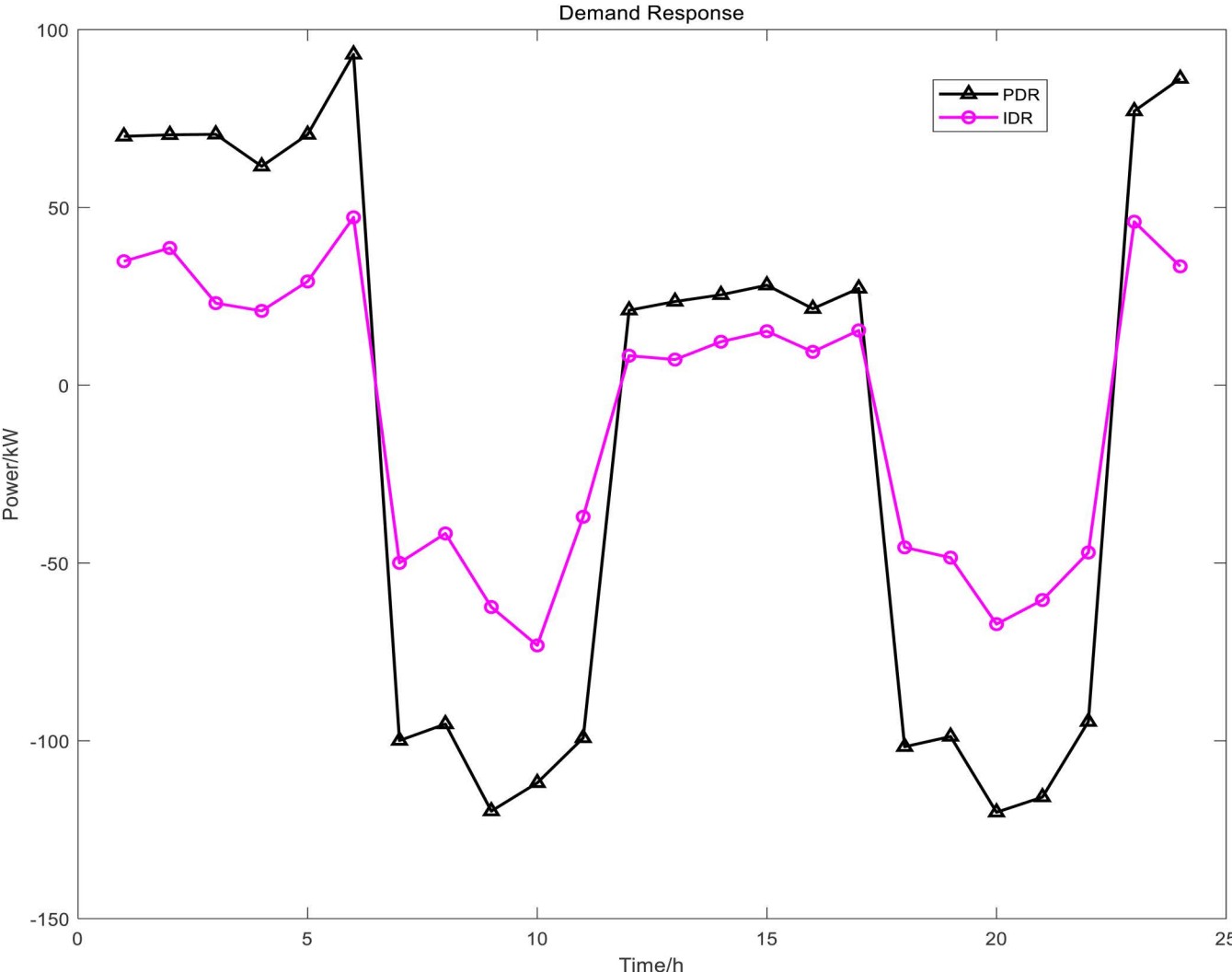

**Fig 7. The Waveforms of PDR and IDR Power.**

Carbon emission trading refers to the government assessing the maximum carbon emissions in a certain area under the conditions of the environment. It is divided it into several carbon emission allowances, and distributed them to enterprises that need to emit carbon emissions through auctions or free allocation. For CCHP-MG established in this article, carbon emission allowances are provided to the system through free allocation and based on the baseline method, with the carbon emission sources being MT, FC, GB, and PG. MT generates electricity and produces heat, GB only produces heat, FC serves as an electricity dispatching role, and PG is considered for upgrading to coal-fired units for thermal power generation. Carbon emission allowances according to the total equivalent calorific value:

$$E_{pt} = \kappa_{MT}^e P_{MTt}^e + \kappa_{MT}^h P_{MTt}^h + \kappa_{GB}^h P_{GBt}^h + \kappa_{FC}^e P_{FCt}^e + \kappa_{PG}^e P_{PGt}^e \tag{39}$$

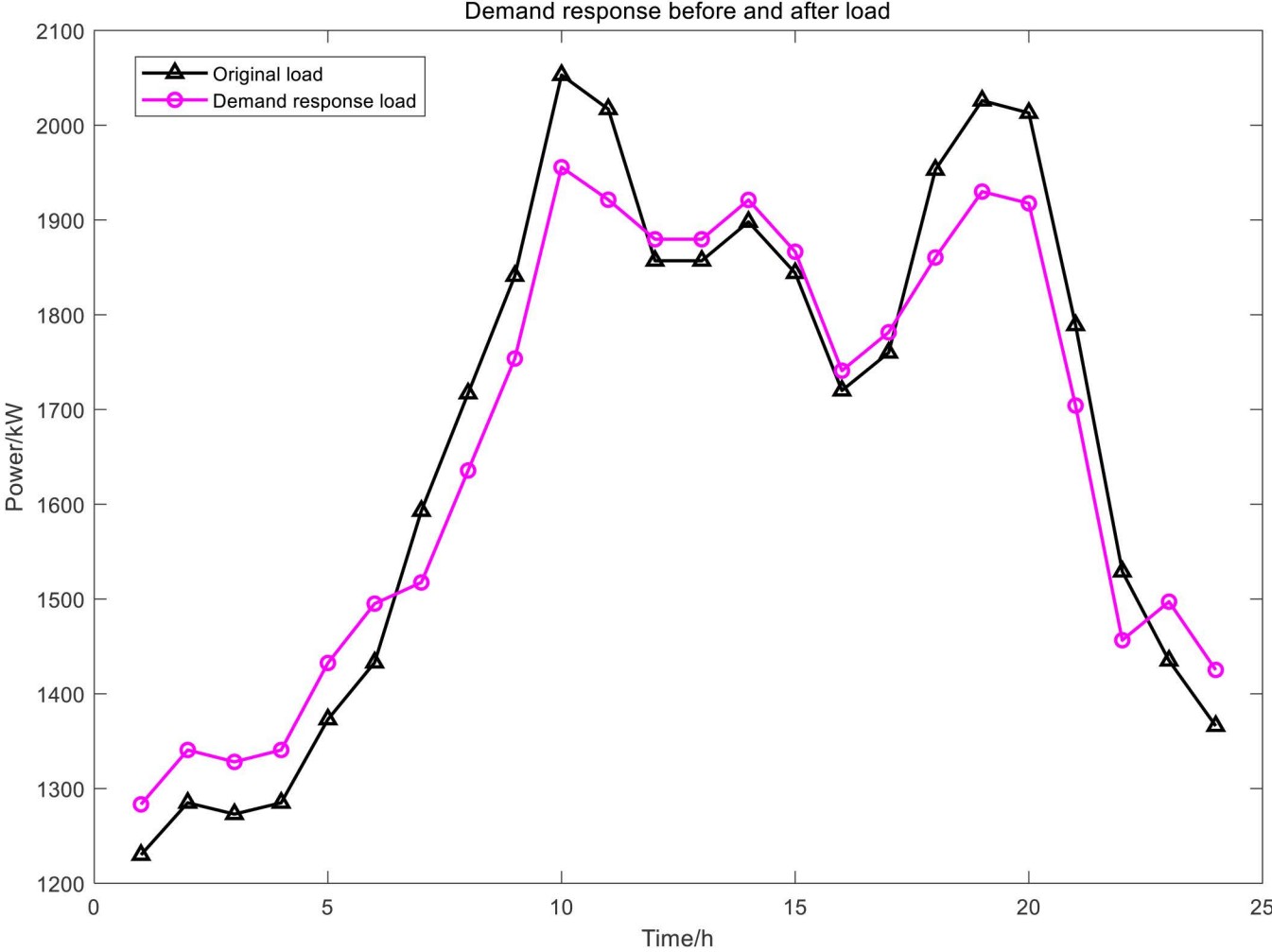

**Fig 8. Electrical load waveform.**

where: $\kappa$ represents the regional unit electricity carbon emission allocation amount, which is determined by the "National Carbon Emission Trading Scheme for Total Quantity and Allocation of Allowances for Power Generation Sector" issued by the Ministry of Ecology and Environment of China in 2024. $\kappa^e_{MT} = \kappa^e_{FC} = 0.3288t/(MW \cdot h)$, $\kappa^e_{PG} = 0.7944t/(MW \cdot h)$, $\kappa^h_{GB} = \kappa^h_{MT} = 0.0533t/GJ$. $E_{apt}$ is the actual emission of CCHP-MG in this paper is similar to Eq. (39), and the carbon emission factors of each system are selected as: $\kappa^e_{aMT} = \kappa^e_{aFC} = 0.4788t/(MW \cdot h)$; $\kappa^e_{aPG} = 0.8944t/(MW \cdot h)$, $\kappa^h_{aGB} = \kappa^h_{aMT} = 0.0682t/GJ$.

In order to encourage the system to actively participate in the carbon trading mechanism, this paper constructs the carbon trading strategy: users can trade carbon emission quotas on their own, if the actual carbon emissions are less than the carbon emission quota, they can sell the remaining carbon emission quotas at market prices to obtain profits. Otherwise, they need to buy the excess carbon emission quota from the market. Therefore, the carbon trading cost in the t period is $C^T_{ca}$, $C_c$ is the trading price.

$$C^T_{ca} = C_c(E_{apt} - E_{pt})$$

(40)

## 4.3 Day-ahead stage

In the day-ahead dispatching, the cost of low-carbon optimization model.

$$F_1 = \min \sum_{T=1}^{24} \left( C_{MT}^T + C_{FC}^T + C_{GB}^T + C_{ES}^T + C_{AIDR}^T + C_{ca}^T + C_{grid}^T \right) \Delta T \tag{41}$$

where $F_1$ represents optimization of the objective. $C_{AIDR}^T$ is the calling cost of A IDR. $C_{gridt}$ and $P_{grid}^T$ are the electricity prices and electricity purchased of the power grid (PG).

$$C_{grid}^T = C_{gridt} P_{grid}^T \tag{42}$$

Electrical power balance constraint:

$$P_{WT}^T + P_{PV}^T + P_{MT}^T + P_{FC}^T + P_{grid}^T - P_{bt,chr}^T + P_{bt,dis}^T = P_{load}^T + P_{ISAC}^T + P_{EB}^T + P_{AIDR} \tag{43}$$

In the equation: $P_{ISAC}^T$ is the power consumption of ISAC in the T period; $P_{WT}^T$ and $P_{PV}^T$ are the output powers of WT and PV, respectively. Cold energy, heat energy and power balance constraints:

$$Q_{MT}^T + Q_a^T + Q_d^T = Q_{load}^T \tag{44}$$

$$H_{MT}^T + H_{EB}^T + H_{GB}^T - H_{tst,chr}^T + H_{tst,dis}^T = H_{load}^T \tag{45}$$

$$P_{grid}^{\min} \leq P_{grid}^T \leq P_{grid}^{\max} \tag{46}$$

where $P_{grid}^{\max}$ and $P_{grid}^{\min}$ are the maximum and minimum quantity of electricity purchased, respectively.

After the day-ahead dispatching stage, the output of the units, energy storage equipment and electricity purchased of PG can be calculated, including: $P_{MT}^T, P_{FC}^T, P_{EB}^T, P_{ISAC}^T, P_{bt,dis}^T, P_{bt,chr}^T, P_{grid}^T, H_{GB}^T, H_{tst,chr}^T, H_{tst,dis}^T$, and the response volume for PDR and Class A IDR is determined. $F_1$ decomposes into two parts, $F_{11}$ and $F_{12}$ are describing as follows:

$$\begin{cases} F_{11} = \sum_{T=1}^{24} \left( C_{MT}^T + C_{GB}^T \right) \Delta T \\ F_{12} = \sum_{T=1}^{24} \left( C_{FC}^T + C_{ES}^T + C_{AIDR}^T + C_{ca}^T + C_{grid}^T \right) \Delta T \end{cases} \tag{47}$$

## 4.4 Intra-day two-layer stage

**4.4.1 Upper-layer rolling optimization model.** In the upper-layer optimization problem, it is necessary to follow the equipment of day-ahead scheduling stage, and adjust the output of each unit and energy storage equipment according to the variation of the cooling and heating load. The objective is $F_{21}$:

$$F_{21} = F_{11} + \min \sum_{t=t_0}^{t_0+30\,\min} (\Delta C_{MT}^t + \Delta C_{GB}^t + \Delta C_{EB}^t + \Delta C_{ISAC}^t) \Delta t \tag{48}$$

$$\begin{cases} \Delta C_{MT}^t = C_{GAS} \frac{\Delta P_{MT}^t}{\eta_{MT} Q_H} + \mu_{MT} \left( \Delta P_{MT}^t \right)^2 \\ \Delta C_{GB}^t = C_{GAS} (\frac{\Delta P_{GB}^t}{\eta_{GB} Q_{H_2}} + \mu_{GB} \left( \Delta P_{GB}^t \right)^2 \\ \Delta C_{EB}^t = \mu_{EB} (\Delta P_{EB}^t)^2 \\ \Delta C_{ISAC}^t = \mu_{ISAC} \left( \Delta P_{ISAC}^t \right)^2 \end{cases} \tag{49}$$

where $\Delta C_{MT}^t$, $\Delta C_{GB}^t$, $\Delta C_{EB}^t$ and $\Delta C_{ISAC}^t$ are the adjustment costs of MT, GB, EB and ISAC, respectively, $t_0$ to $t_0+30$ min is the control time domain for heat and cold scheduling; $\Delta P_{MT}^t$, $\Delta P_{GB}^t$, $\Delta P_{EB}^t$ and $\Delta P_{ISAC}^t$ are the adjusted power of MT, GB, EB, and ISAC, $\mu_{MT}$, $\mu_{GB}$, $\mu_{EB}$ and $\mu_{ISAC}$ are the penalty costs for adjusting MT, GB, EB, and ISAC, respectively.

The cold and heat energy of intra-day scheduling stage must also meet the following constraints:

$$\begin{cases} Q_{MT}^t + Q_a^t + Q_d^t = Q_{load}^t \\ H_{MT}^t + H_{EB}^t + H_{GB}^t - H_{tst,chr}^t + H_{tst,dis}^t = H_{load}^t \end{cases} \tag{50}$$

The constraint cold and heat energy of the intra-day stage can be referred to the day-ahead stage. In addition, the fine-tuning change of MT does not exceed 5% of the maximum value.

**4.4.2 Lower-layer rolling optimization model.** In the lower-layer optimization objective function is Eq. (51). Revision of the day-ahead schedule is based on fluctuations in renewable energy sources and power changes in electrical loads.

$$F_{22} = F_{12} + \min \sum_{t=t_0}^{t_0+15\,\text{min}} (\Delta C_{grid}^t + \Delta C_{FC}^t + \Delta C_{ES}^t + C_{BIDR}^t + \Delta C_{ca}^t)\Delta t \tag{51}$$

$$\begin{cases} \Delta C_{grid}^t = C_{gridt}\Delta P_{grid}^t + \mu_{grid}\left(\Delta P_{grid}^t\right)^2 \\ \Delta C_{FC}^t = C_{GAS}(\frac{\Delta P_{FC}^t}{\eta_{FC}Q_H}) + \mu_{FC}(\Delta P_{FC}^t)^2 \\ \Delta C_{ES}^t = \mu_{ES}[(\Delta P_{bt,dis}^t)^2 + (\Delta P_{bt,chr}^t)^2] \end{cases} \tag{52}$$

where $\Delta C_{grid}^t$, $\Delta C_{FC}^t$ and $\Delta C_{ES}^t$ are the adjustment costs of the PG, FC and ES, respectively. $t_0$ to $t_0+15$ min is the control time domain of power scheduling; $\Delta P_{grid}^t$, $\Delta P_{FC}^t$, $\Delta P_{bt,dis}^t$ and $\Delta P_{bt,chr}^t$ are the adjustment power of PG, FC and ES, respectively. $\mu_{grid}$, $\mu_{FC}$ and $\mu_{ES}$ are the penalty costs for adjusting PG, FC and ES, respectively. $C_{BIDR}^t$ and $\Delta C_{ca}^t$ are the invocation cost and the cost of intra-day carbon emissions.

The constraint for intra-day electricity dispatching is similar to the day-ahead electricity dispatching. Additionally, it is necessary to ensure that the change in capacity of ES and PG do not exceed 5% of their maximum amount, respectively.

$$P_{WT}^t + P_{PV}^t + P_{MT}^t + P_{FC}^t + P_{grid}^t - P_{bt,chr}^t + P_{bt,dis}^t = P_{load}^t + P_{ISAC}^t + P_{EB}^t + P_{BIDR} \tag{53}$$

## 4.5 Real-time stage

The real-time model focus on the discrepancy between the forecast output and actual the of WT and PV power, the real-time fluctuation of load. This stage uses ES, FC and class C IDR to suppress fluctuations, improve the problems of load loss and WT and PV power reductions, and construct a low-carbon scheduling model. The objective function is $F_3$:

$$F_3 = F_{21} + F_{22} + \min \sum_{\tau=\tau_0}^{\tau_0+5\,\text{min}} (\Delta C_{grid}^\tau + \Delta C_{FC}^\tau + \Delta C_{ES}^\tau + C_{CIDR}^\tau + \Delta C_{ca}^\tau)\Delta\tau \tag{54}$$

where $\Delta C_{grid}^\tau$, $\Delta C_{FC}^\tau$ and $\Delta C_{ES}^\tau$ are the adjustment costs of the PG, FC and ES, $C_{CIDR}^\tau$ and $\Delta C_{ca}^\tau$ are the invocation cost and the cost of real-time fine-tuning carbon emissions. $\tau_0$ to $\tau_0 + 5min$ is the control time domain of power scheduling. The remaining formulas and constraints are similar to intra-day scheduling.

## 5 Example analysis

This study employs operational dataset from an industrial park in Jiangsu Province, China, covering the period from May to July over a 24-year span, with the original data being provided and validated by Jiangsu Pinggao Tai Shida Electric Co.

LTD under a confidentiality agreement. The simulation is conducted in MATLAB R2023a and solved by Gurobi solver, with detailed equipment parameters provided in Table 5. In order to validate the reasonableness of the proposed model, we compared and analyzed four scenarios.

Scenario 1: Considering DR under the carbon trading mechanism.

Scenario 2: Only considering the carbon trading mechanism.

Scenario 3: Only considering DR.

Scenario 4: Without considering the carbon trading mechanism and without considering DR.

The cost of each scenario is shown in Table 6.

Table 6 shows that compared to Scenario 4, Scenario 1 had an 18% lower cost of carbon emissions, and the actual carbon emissions were reduced by 17683.08 kg. The reason is that Scenario 1 considered the carbon trading market, which allowed CCHP-MG to have an initial carbon emission quota that can offset part of the carbon emission cost, while Scenario 4 needed to consider the all cost of the carbon emission. Compared to Scenario 4, the energy purchase cost of Scenario 3 reduced by 10%, which was due to considering DR to reduce peak-time electrical loads and increased

**Table 5. Parameters of devices.**

| Device type | Parameters | Value |
|---|---|---|
| MT1/MT2 | $P_{MT1}^{min}/P_{MT2}^{min}$(kW) | 15/15 |
| | $P_{MT1}^{max}/P_{MT2}^{max}$(kW) | 600/400 |
| | $\eta_{MT}$ | 0.4 |
| | $C_{OP,h}/C_{OP,c}$ | 1.5/0.95 |
| | $\eta_h/\eta_c$ | 0.5/0.45 |
| FC | $P_{FC}^{min}/P_{FC}^{max}$(kW) | 15/350 |
| | $\eta_{FC}$ | 0.7 |
| ES | $S_{SOC}^{min}/S_{SOC}^{max}$(kW·h) | 400/1600 |
| | $\eta_{bt,dis}/\eta_{bt,chr}$ | 0.9/0.95 |
| | $P_{bt,dis}^{max}/P_{bt,chr}^{max}$(kW) | 150/200 |
| EB | $P_{EB}^{max}$(kW) | 900 |
| | $\eta_{EB}$ | 0.9 |
| GB | $H_{GB}^{min}/H_{GB}^{max}$(kW) | 0/1400 |
| | $\eta_{GB}$ | 0.85 |
| HS | $S_{tst}^{min}/S_{tst}^{max}$(kW·h) | 200/1500 |
| | $H_{tst,dis}^{min}/H_{tst,chr}^{max}$(kW) | 150/200 |
| | $\eta_{tst,dis}/\eta_{tst,chr}$ | 0.90/0.95 |
| ISAC | $Q_a^{min}/Q_a^{max}$(kW) | 50/400 |
| | $Q_d^{max}$(kW) | 150 |
| | $\eta_{ice,dis}/\eta_{ice,chr}$ | 0.90/0.95 |

**Table 6. Carbon emissions and costs of four scenarios.**

| Scenario | Total Run Cost/yuan | Energy Purchase Cost/yuan | Carbon trading Cost/yuan | Carbon emissions/kg |
|---|---|---|---|---|
| 1 | 48510.51 | 38549.31 | 1069.65 | 73083.60 |
| 2 | 49335.42 | 39241.56 | 1115.79 | 76207.95 |
| 3 | 54422.37 | 37060.92 | 8364.75 | 84492.42 |
| 4 | 55586.19 | 41297.25 | 8273.04 | 90766.68 |

valley-time electrical loads, thereby enabling the system to choose more economical ways to buy energy. Compared with Scenario 1 and Scenario 2, Scenario 3 had a higher operational cost, lower energy purchase cost, and higher carbon trading cost and actual carbon emissions. The carbon trading mechanism plays a significant role in promoting carbon emission reduction and energy conservation. The total operation cost, carbon trading cost, energy purchase cost, and actual carbon emissions of scenario 1 were all less than those of Scenario 2. This is since considering DR under the carbon trading mechanism, on the one hand, shifted part of the load from the high electricity price time domain to the low electricity price time domain and reduced part of the load, and on the other hand, realized the mutual substitution of electricity and heat on the user side and smoothed the load curve. CCHP-MG selected a more economical and with less carbon emissions by comparing the electricity and gas purchase costs and the output of MT, FC, and PG electricity purchased in different periods, effectively coordinating the operation economy and low-carbon nature of the system.

In Scenario 1, the day-ahead scheduling power balance diagram, heat energy balance diagram, and cold energy balance diagram are shown in Fig 9–11. In this paper, MT selected two devices, MT1 and MT2, to participate in the day-ahead scheduling. During the 22:00–07:00 electricity price in the valley period, both the electrical load and the cold load were in the valley, the output of renewable energy was gradually increasing, the heat load was mainly borne by GB and EB, the electrical load was mainly borne by the power grid purchase, and the cold load was mainly borne by the ISAC refrigeration power. During the peak period of electricity price from 07:00 to 11:00, with the gradual increase of electric

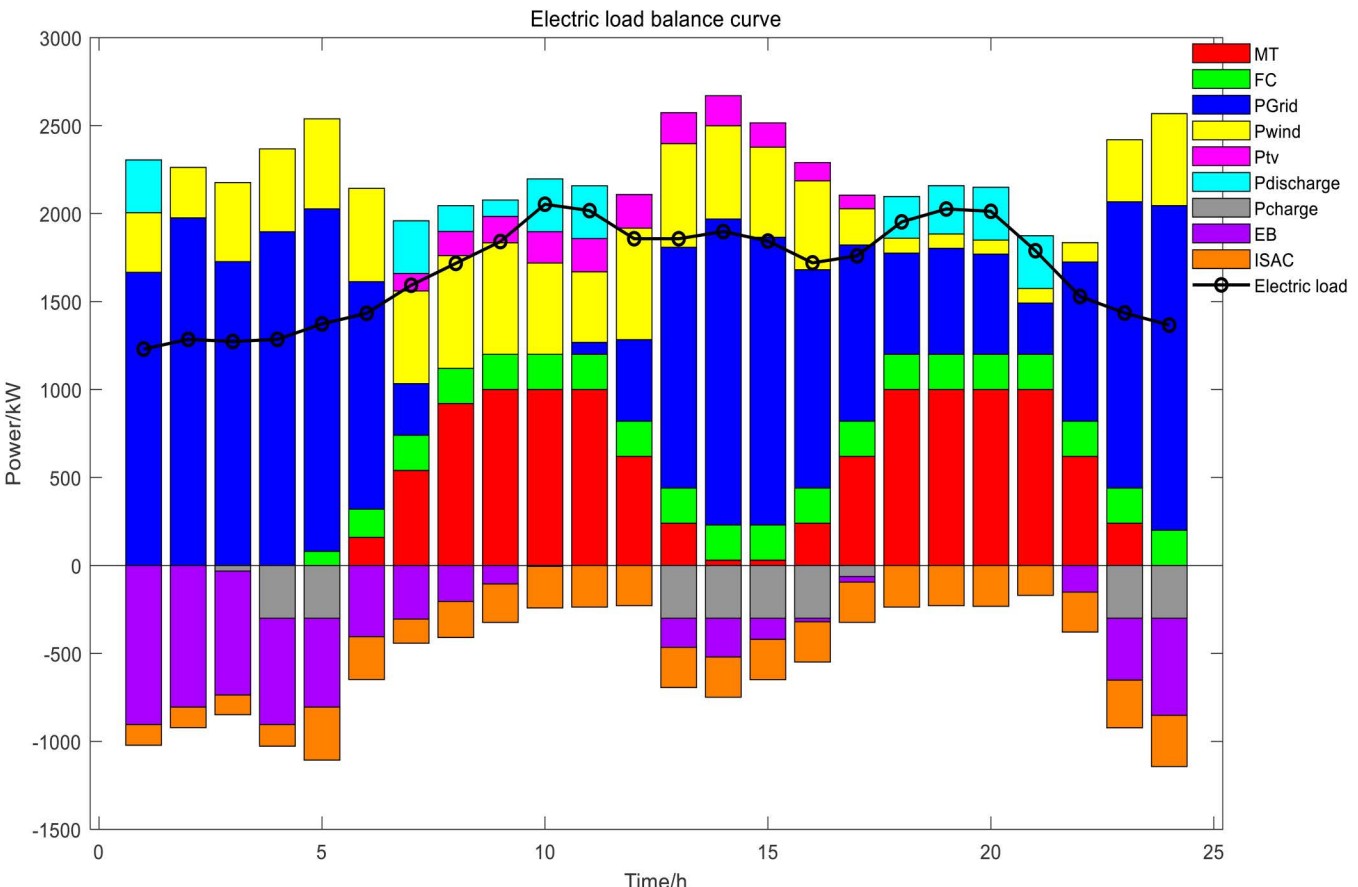

**Fig 9. Day-ahead scheduling power balance diagram.**

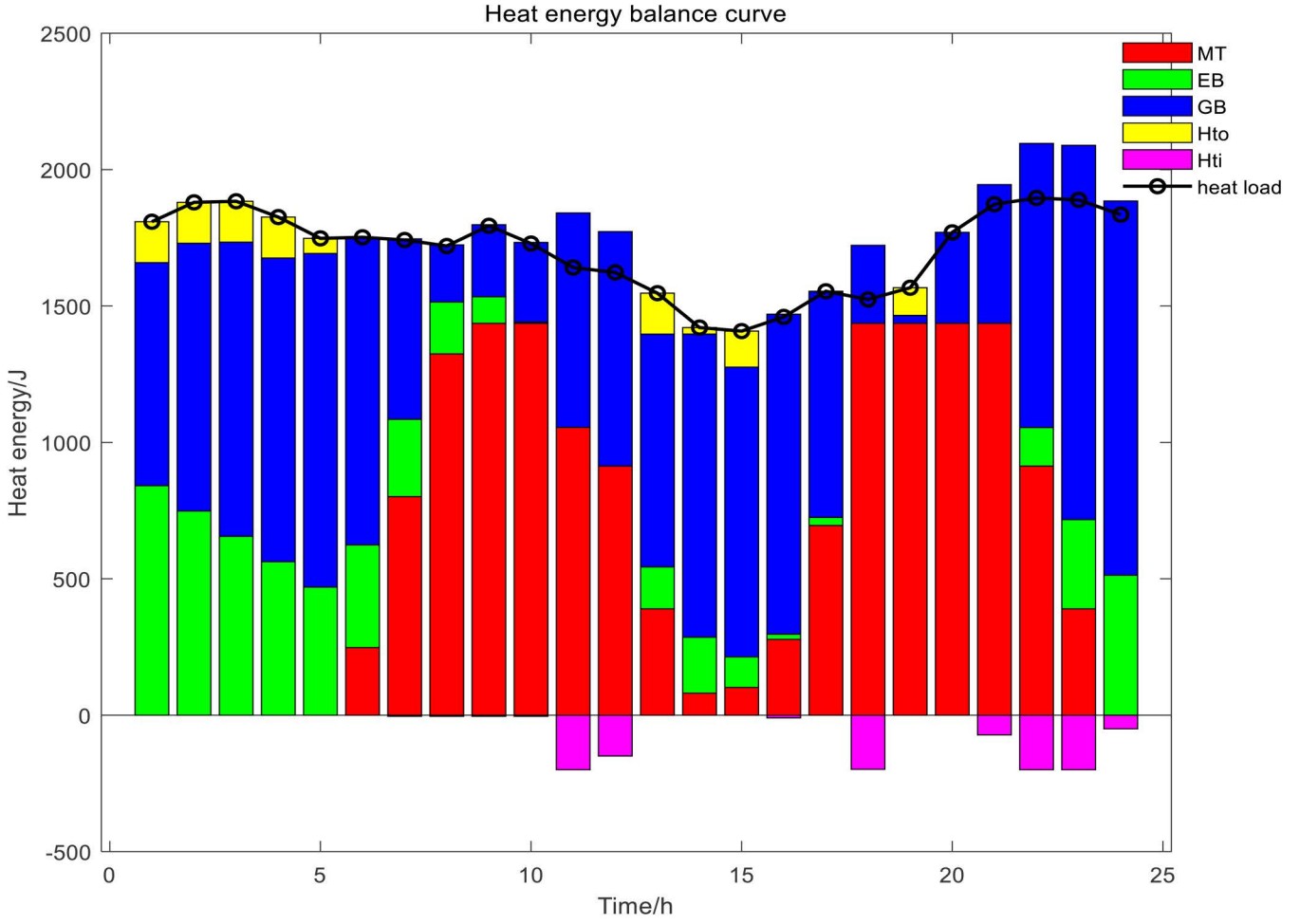

**Fig 10. Day-ahead scheduling heat energy balance diagram.**

load and cold load, the output of MT and FC continued to increase, and reached full load operation for the same time, the output of renewable energy also increased at this time, electrical load of the system reduced dependence on the external grid at this time, ES and ISAC used less electricity power, excess heat was stored by the HS and the cold storage tank began to release cold energy. During the flat period of electricity price from 11:00 to 17:00, the electric load gradually decreased, meanwhile, ES gradually stored energy, MT gradually reduced the output to provide less heat and cold energy, the power grid purchases increased, HS began to release heat energy, GB heat production increased, and ISAC refrigeration power increased. From 17:00 to 22:00, the electricity price reached the peak period again, and the working condition of the system was basically the same as that of the 06:00–11:00 period, but considering the low WT and PV power output in the current period, the discharge frequency of ES was higher than that of the first peak electricity price. The power and energy changes of each unit and energy storage device of day-ahead scheduling are shown in Fig 12 Day-ahead scheduling power and energy changes.

The optimization results of intra-day upper layer cooling and heating load scheduling are shown in Fig 13. In this paper, MT selected two devices, only MT2 participates in the scheduling of intra-day cooling and heating load fluctuations, and MT1 does not participate. The scheduling of cold and heat energy in intra-day scheduling was based on a time scale of

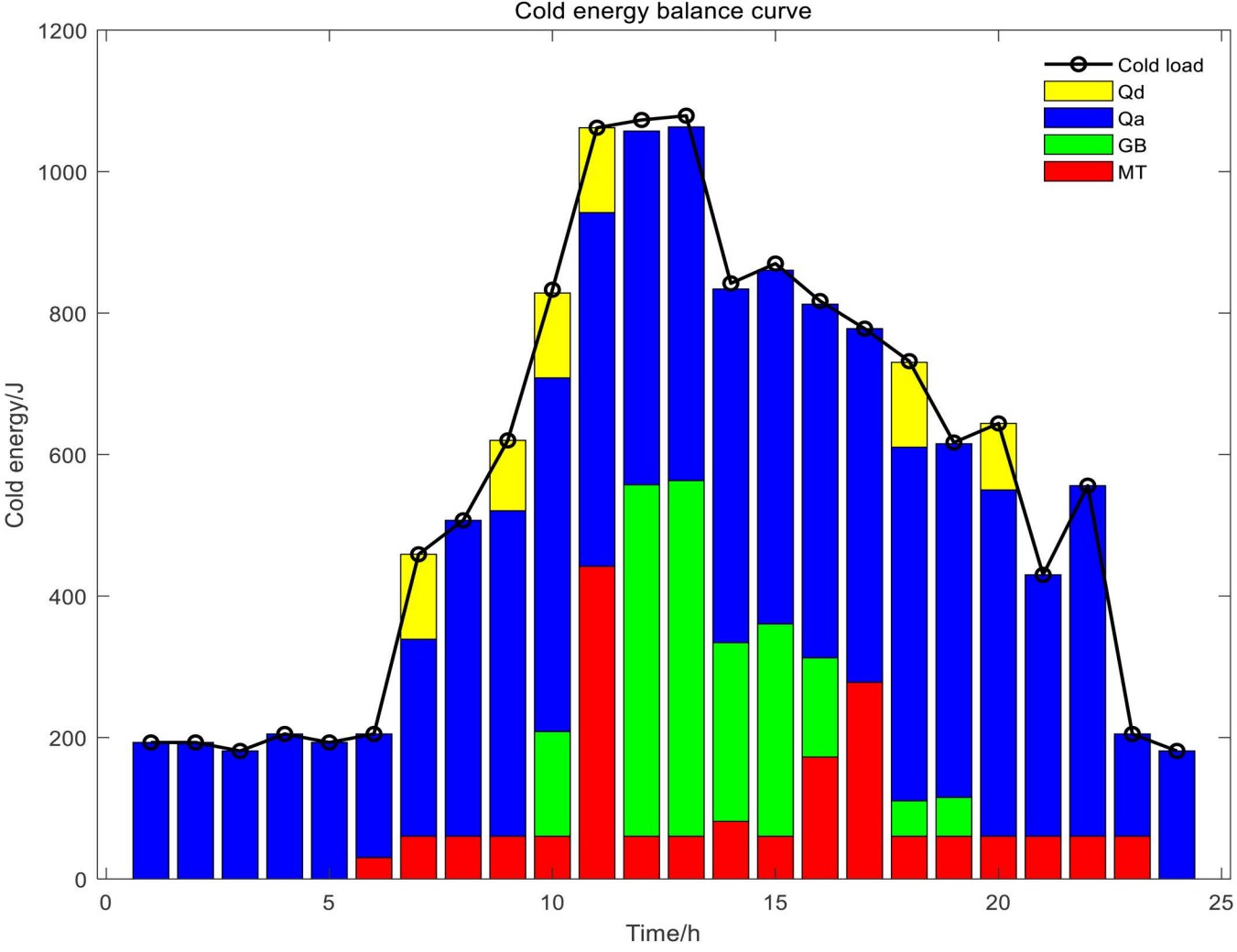

**Fig 11. Day-ahead scheduling cold energy balance diagram.**

30 min, which is more accurate than the day-ahead prediction, so the equipment output was corrected. The optimization results of intra-day lower layer power load scheduling are shown in Fig 14. According to the results of the upper layer cold and heat energy scheduling and the fluctuation of the electric power load, the day-ahead electric energy scheduling plan was adjusted. At this time, the FC, ES output and power grid purchase were adjusted to meet the class B IDR load invocation plan and the scheduling of intra-day power load fluctuations, and the FC had sufficient reserve margin. The scheduling of power energy in intra-day dispatch was based on a time scale of 15 min.

The real-time scheduling used 5 min time scale, so it expanded to 288 data points in real-time scheduling. According to the intra-day dispatch results, the error between the forecast and actual WT and PV power. The fluctuation of electric load, and the load calling plan of class C IDR to adjust FC, ES and power grid purchase, and the real-time stage optimization are shown in the Fig 15.

The DR was evident from the results of the day-ahead, in-day and real-time scheduling, as shown in Fig 16. It can be seen from the day-ahead scheduling that the basic trend of peak shaving and valley filling for PDR and class A IDR was

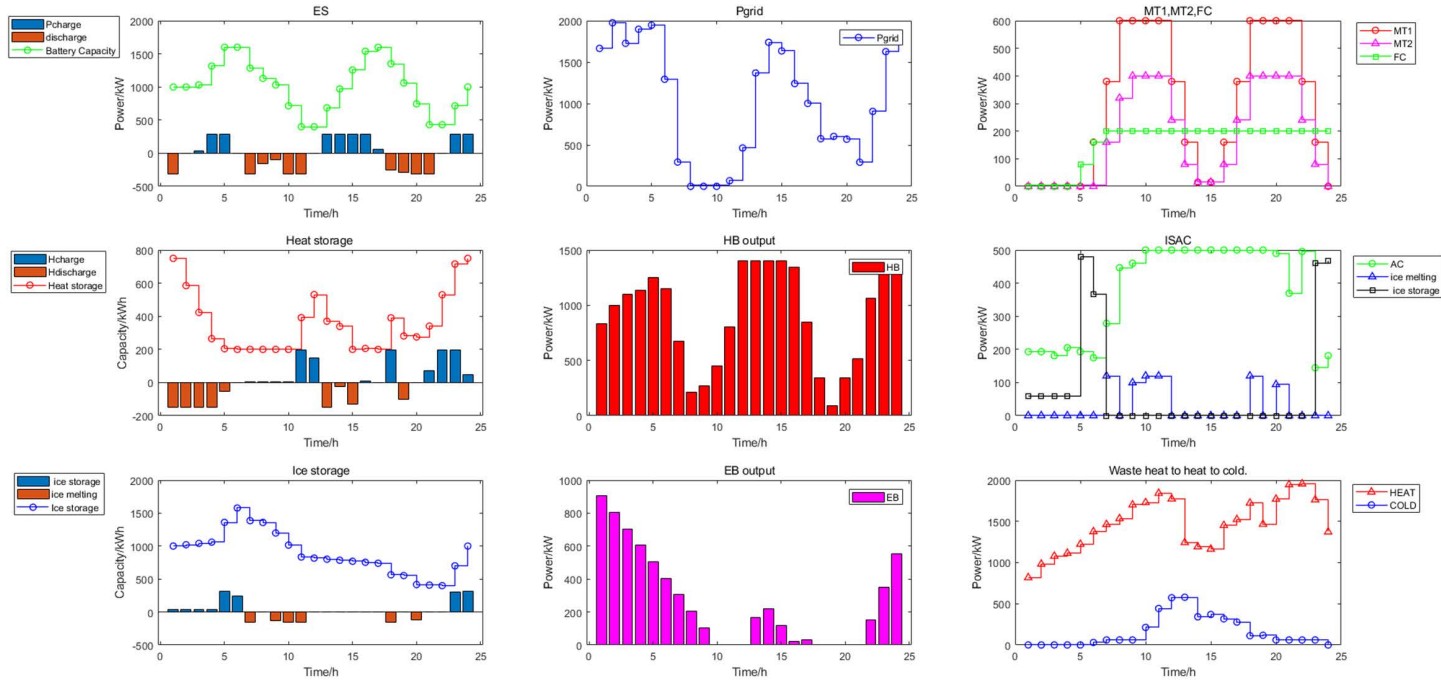

**Fig 12. Day-ahead scheduling power and energy changes.**

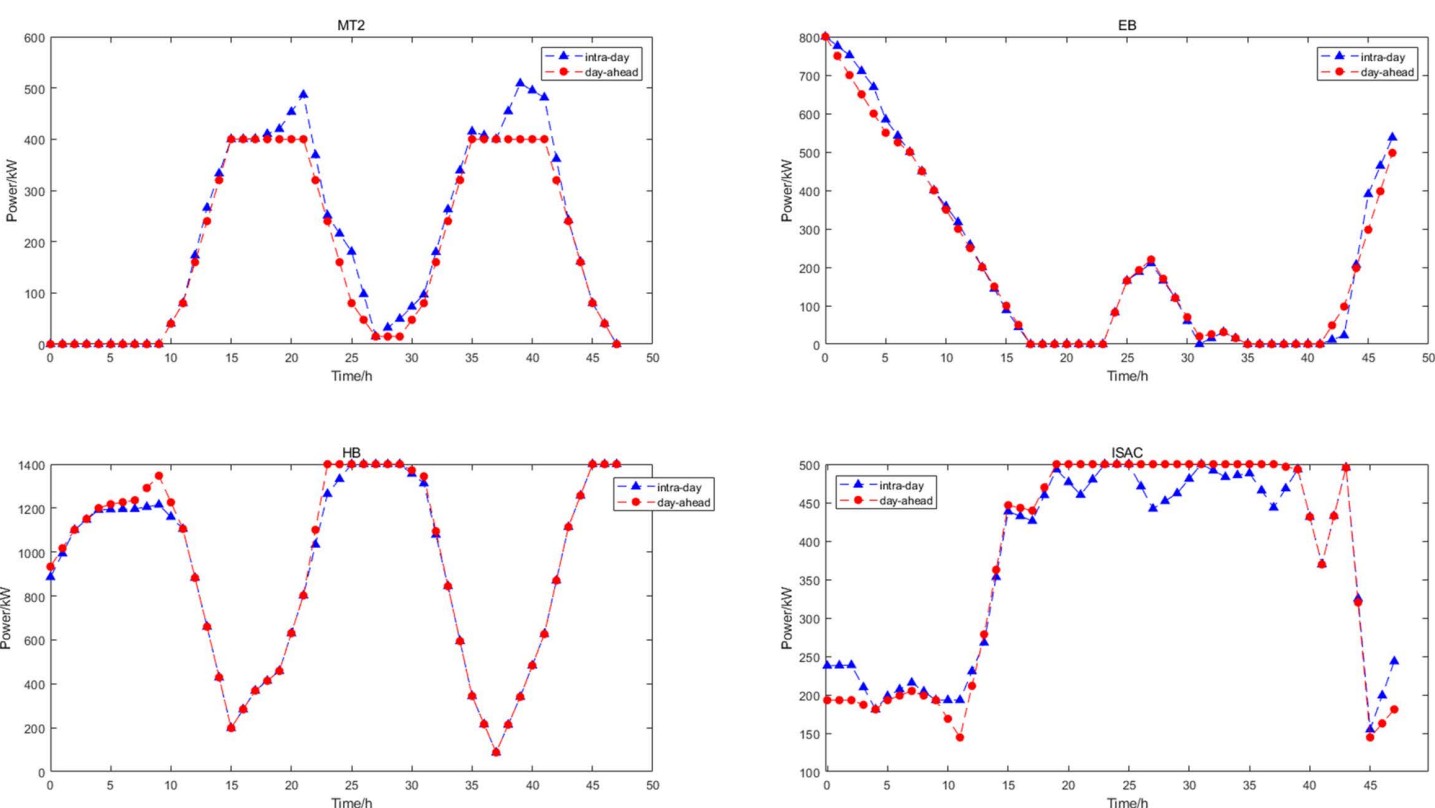

**Fig 13. Intra-day upper cooling and heating load scheduling.**

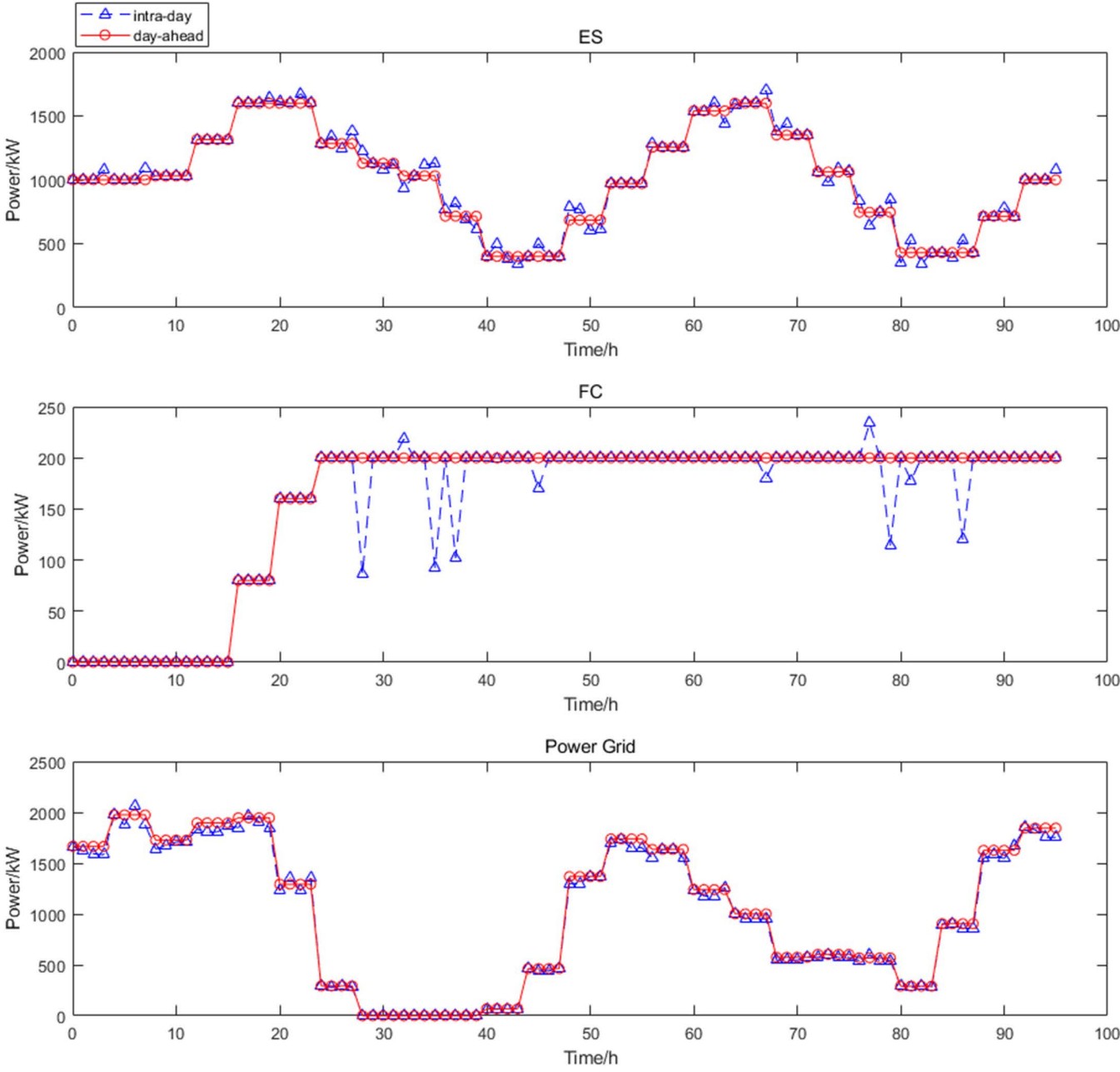

**Fig 14. Intra-day lower power scheduling.**

to increase load during the valley and flat period of electricity prices and to reduced load during the two peak periods of electricity price. During the intra-day lower layer scheduling, class B IDR was called more frequently during the daytime and the changes were more dramatic, and mainly fluctuated greatly during the peak load period in the daytime, class B IDR and FC, ES output, as well as power grid purchase, to smooth intra-day fluctuations. In real-time scheduling, class C IDR in 24h a day, although the magnitude of fluctuations was small, the changes were dramatic, mainly used to smooth

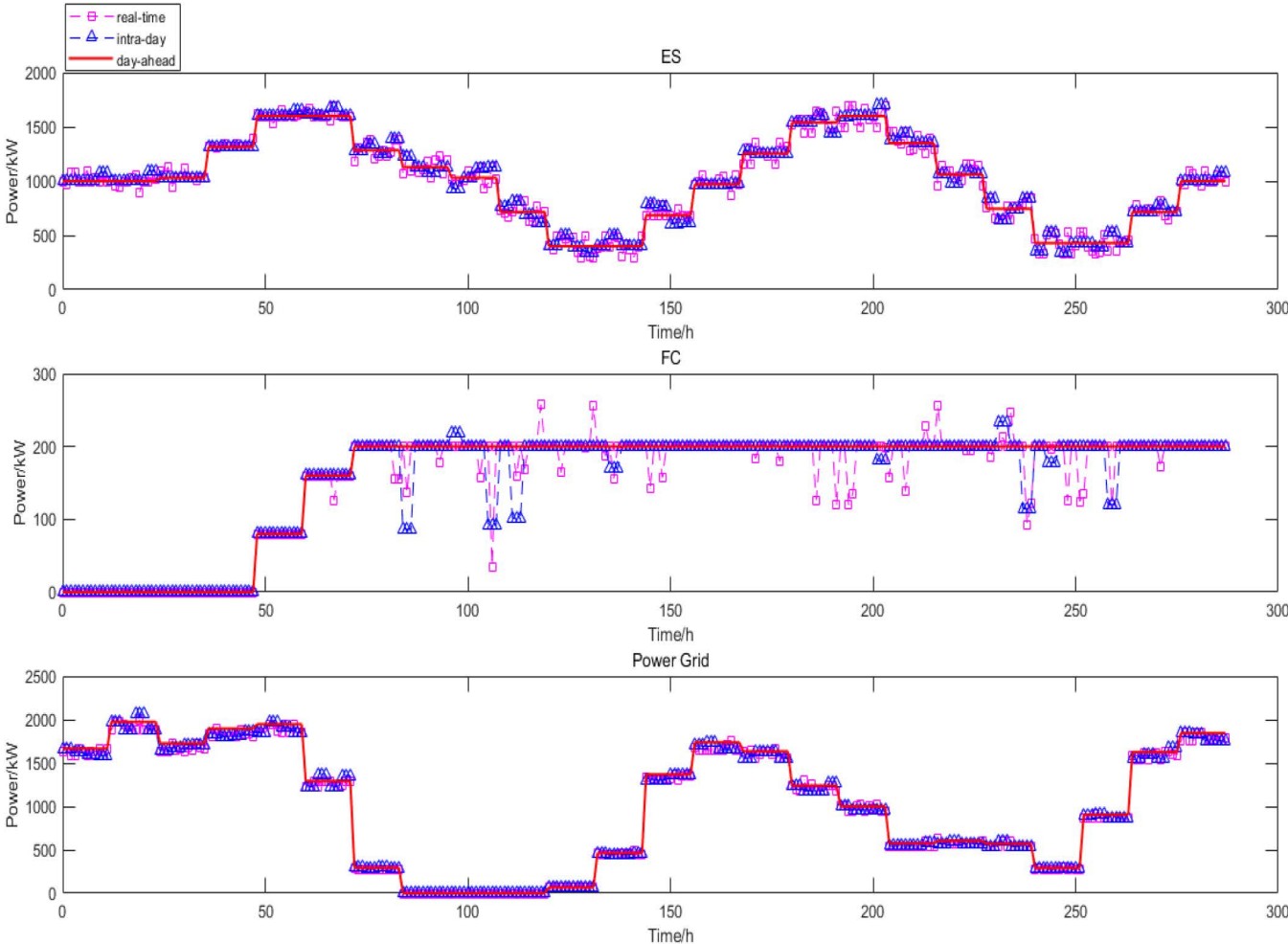

**Fig 15. Real-time power scheduling.**

out the fluctuations of WT and PV power under a very short time scale, similar to the intra-day scheduling worked with the FC, ES and power grid purchase.

## 6 Conclusions and future work

This study investigates CCHP-MG integrated with renewable distributed generation. Considering the temporal correlation and complementarity of cooling, heating, and power loads, we propose a multi-time scale scheduling model incorporating DR and a carbon trading mechanism. We developed a wind turbine (WT) and photovoltaic (PV) output forecasting model. The main findings are as follows:

(1) We developed a CNN-ATT-BiLSTM model by integrating an attention mechanism into a combined CNN and BiLSTM network for WT and PV power forecasting. This hybrid model demonstrated superior prediction accuracy compared to individual CNN or BiLSTM models.

(2) In this study, MPC approach is adopted to optimize the operation of a comprehensive energy system across day-ahead to real-time scheduling horizons. The proposed bi-level optimization framework facilitates multi-timescale

**Fig 16. The results of DR scheduling.**

coordination of CCHP-MG flows. The method demonstrates superior capability in mitigating supply-demand fluctuations while ensuring operational stability for both the combined cooling, heating, and CCHP-MG and the external PG.

(3) Case studies evaluating four operational scenarios with DR and carbon trading demonstrated that the multi-time scale model simultaneously satisfies user demands for cooling, heating, and power while significantly reducing supply-demand fluctuations. Energy output analyses of system components and storage devices confirmed that the proposed approach achieves economical, environmentally friendly, and stable CCHP-MG operation.

To further advance this field, future research could prioritize the following directions:

(1) Computational Efficiency for High-Dimensional Systems – Future studies should focus on developing lightweight MPC variants to enable real-time control in large-scale systems, including distributed energy networks and multi-agent cooperative systems. Further exploration of edge computing integration could enhance computational scalability.

(2) Robustness Under Uncertainty – Enhancing the current framework to address model mismatch, non-Gaussian disturbances, and adversarial scenarios—potentially through distributionally robust optimization or adaptive learning techniques—would significantly improve practical applicability in uncertain environments.

(3) Dynamic Granularity Adaptation – The development of self-adaptive time-scale partitioning algorithms, such as those leveraging event-triggered mechanisms or digital twin feedback, could dynamically optimize scheduling granularity in response to renewable forecasting errors and demand fluctuations.

(4) Equipment Dynamics and Robustness Verification – Future investigations should systematically examine the dynamic interactions between equipment response characteristics and system-level stability, such as comprehensive parameter sensitivity analysis of device response rates and development of integrated robustness testing frameworks capable of evaluating the coupled dynamics-stability relationships in multi-energy scenarios.

These research directions aim to bridge current theoretical and practical gaps while enhancing the applicability of multi-timescale optimal scheduling methodologies for integrated energy systems in complex real-world scenarios.

## Author contributions

**Conceptualization:** Jue Wang, Dengfeng Zhang.

**Data curation:** Jue Wang, Zhiwei Cheng, Dejun Lu.

**Methodology:** Jue Wang, Dengfeng Zhang.

**Software:** Jue Wang, Zhiwei Cheng.

**Writing – original draft:** Jue Wang, Dengfeng Zhang.

**Writing – review & editing:** Zhiwei Cheng, Dejun Lu, Mingxiang Zhu.

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
