## [Decision Letter · Decision Letter 0]

Dear Dr. Zhang,

Thank you for submitting your manuscript to PLOS ONE. After careful consideration, we feel that it has merit but does not fully meet PLOS ONE’s publication criteria as it currently stands. Therefore, we invite you to submit a revised version of the manuscript that addresses the points raised during the review process.

**ACADEMIC EDITOR: Please revise**

We look forward to receiving your revised manuscript.

Kind regards,

Zhengmao Li

Academic Editor

PLOS ONE

“This work was supported by the National Natural Science Foundation of China under Grant: 62333010.”

6. We note that Figure 1 in your submission contain copyrighted images. All PLOS content is published under the Creative Commons Attribution License (CC BY 4.0), which means that the manuscript, images, and Supporting Information files will be freely available online, and any third party is permitted to access, download, copy, distribute, and use these materials in any way, even commercially, with proper attribution. For more information, see our copyright guidelines: http://journals.plos.org/plosone/s/licenses-and-copyright.

Reviewers' comments:

Reviewer's Responses to Questions

**Comments to the Author**

1. Is the manuscript technically sound, and do the data support the conclusions?

Reviewer #1: Yes

Reviewer #2: Yes

2. Has the statistical analysis been performed appropriately and rigorously?

Reviewer #1: N/A

Reviewer #2: Yes

3. Have the authors made all data underlying the findings in their manuscript fully available?

Reviewer #1: No

Reviewer #2: Yes

4. Is the manuscript presented in an intelligible fashion and written in standard English?

Reviewer #1: Yes

Reviewer #2: Yes

Reviewer #1: This paper proposes a Multi-Time Scale Rolling Optimization Framework to address high renewable penetration, load demand uncertainty, and heterogeneous response timescales across energy subsystems. The proposed approach appears to be technically feasible and has potential practical value. I have the following comments and suggestions for improvement:

1. Since intra-day forecasting is introduced, short-term rolling optimization may suffer from myopic behavior compared to global optimization, which could negatively impact the overall cost. Can your method address this issue? Furthermore, given the real-time adjustment of schedules, the absence of anticipatory constraints might result in infeasibility in later stages. Would it be necessary to introduce slack variables to ensure feasible dispatch across stages?

2. The solution method for the proposed model is not clearly presented. I recommend elaborating on the algorithmic aspects and analyzing the scalability of the approach—namely, to what extent can your method handle large-scale problems while still meeting the real-time requirements?

3.Does the proposed mechanism support distributed implementation?

4. How accurate is the proposed forecasting model? A comparative analysis with other existing forecasting methods is suggested to highlight its effectiveness.

5. It is recommended to include a brief discussion on future work in the conclusion to provide a forward-looking perspective.

Reviewer #2: This study proposes a multi-time-scale rolling optimization framework that combines the carbon trading mechanism and Demand Response (DR) to improve the low-carbon operation efficiency of a Combined Cooling, Heating and Power Microgrid (CCHP-MG). My comments are as follows:

The CNN-ATT-BiLSTM model proposed in the paper is only compared with CNN and BiLSTM, and other mainstream prediction models (such as Transformer, GRU, etc.) or hybrid models (such as CNN-LSTM) are not included as baselines, making it difficult to comprehensively demonstrate its superiority.

The data only mentions "Data will be made available on request", but the specific sources, time range, or preprocessing methods are not clearly stated.

The scenario comparison only shows the results of the total cost and carbon emissions, lacking a quantitative analysis of the specific scheduling strategies (such as the start-stop timing of equipment and the charging and discharging modes of energy storage). It is recommended to supplement dynamic diagrams of the scheduling process during typical periods (such as the peak and valley electricity price periods) and explain how DR and the carbon trading mechanism collaborate to optimize the output of equipment.

The literature review part does not fully cover relevant studies in recent years (such as multi-time-scale scheduling based on reinforcement learning, integration of hydrogen energy storage, etc.), and there is a lack of critical analysis of the disadvantages of existing methods. It is recommended to expand the scope of the literature and clarify the unique contributions of the model in this paper compared with similar works.

It is recommended to supplement the following literature (not mine):

Distributed Hybrid-Triggered Observer-Based Secondary Control of Multi-Bus DC Microgrids Over Directed Networks

DOI: 10.1109/TCSI.2024.3523339

Resilient Frequency Regulation for Microgrids Under Phasor Measurement Unit Faults and Communication Intermittency

DOI: 10.1109/TII.2024.3495785

V2Sim: An Open-Source Microscopic V2G Simulation Platform in Urban Power and Transportation Network, IEEE Transactions on Smart Grid

DOI: 10.1109/TSG.2024.3417294

A tri-level demand response framework for EVCS flexibility enhancement in coupled power and transportation networks, IEEE Transactions on Smart Grid

DOI: 10.1109/TSG.2025.3560976

The paper assumes that the response rates of all equipment are fixed (such as the ramp rate of the Micro-Turbine (MT) and the charging and discharging efficiency of energy storage), but does not discuss parameter sensitivity or conduct robustness tests (such as the system stability when the prediction errors of wind and solar power increase).

It is necessary to unify the format of the charts, supplement the necessary annotations, and check the continuity of the formula numbering.

**Do you want your identity to be public for this peer review?** For information about this choice, including consent withdrawal, please see our Privacy Policy

Reviewer #1: No

Reviewer #2: No

---

## [Author Response · Author response to Decision Letter 1]

10 Jun 2025

Dear editor,

Thank you for your letter and the reviewers’ comments on our manuscript entitled “A multi-time scale rolling optimization framework for low-carbon operation of CCHP microgrids with demand response integration”. We would like to thank the anonymous referees for their kind comments and valuable suggestions. We have studied the comments carefully and made corrections point-by-point. Revised portions are marked in red in the manuscript or in the supplementary material. The main corrections in the manuscript and the responses to reviewers’ comments are as follows.

Thank you for taking time out of your busy schedule to read our research work again. Your valuable suggestions are very helpful for the revision of our paper. We have read your opinions carefully and revised our manuscript. We hope this revision can meet your standards. Here are the point-to-point responses.

Response: We have revised the manuscript as PLOS ONE's style requirements, including those for file naming.

Response: We provided the correct grant numbers for our study in the ‘Funding Information’ section.

3. Thank you for stating the following in the Acknowledgments Section of your manuscript: “This work was supported by the National Natural Science Foundation of China under Grant: 62333010.” We note that you have provided funding information that is currently declared in your Funding Statement. However, funding information should not appear in the Acknowledgments section or other areas of your manuscript. We will only publish funding information present in the Funding Statement section of the online submission form.

Response: The funding acknowledgments were omitted from the main text and instead provided through the online submission system.

4. When completing the data availability statement of the submission form, you indicated that you will make your data available on acceptance.

Response: We revised our statement. The original data were collected from an industrial park in Taizhou, Jiangsu Province, and provided by Company A under a non-disclosure agreement. To protect proprietary business information and comply with contractual obligations, the dataset has been anonymized and processed to remove sensitive details while retaining analytical utility.

5. PLOS requires an ORCID iD for the corresponding author in Editorial Manager on papers submitted after December 6th, 2016. Please ensure that you have an ORCID iD and that it is validated in Editorial Manager.

Response: We have an ORCID iD.

6. We note that Figure 1 in your submission contain copyrighted images.

Response: We update the Fig 1.

Reviewer #1:

Thank you for taking time out of your busy schedule to read our research work again. Your valuable suggestions are very helpful for the revision of our paper. We have read your opinions carefully and revised our manuscript. We hope this revision can meet your standards. Here are the point-to-point responses.

1. Since intra-day forecasting is introduced, short-term rolling optimization may suffer from myopic behavior compared to global optimization, which could negatively impact the overall cost. Can your method address this issue? Furthermore, given the real-time adjustment of schedules, the absence of anticipatory constraints might result in infeasibility in later stages. Would it be necessary to introduce slack variables to ensure feasible dispatch across stages?

Response: We sincerely appreciate your insightful feedback. Your observation regarding the potential myopic behavior of short-term rolling optimization and the risk of infeasibility due to real-time adjustments is indeed critical. Upon reflection, we recognize that our may not have fully addressed these challenges, and we appreciate the opportunity to clarify our methodology. We have modified the manuscript and added the following sentence to last paragraph of 4.1 section:

2. The solution method for the proposed model is not clearly presented. I recommend elaborating on the algorithmic aspects and analyzing the scalability of the approach—namely, to what extent can your method handle large-scale problems while still meeting the real-time requirements?

Response: Thanks for your good suggestion. We have added the solution method for the proposed model. We also have modified the manuscript and added the following sentence to paragraph 2 of the 4.1 section:

3. Does the proposed mechanism support distributed implementation?

Response: The proposed mechanism in this study is inherently centralized and does not accommodate a distributed implementation framework. The Model Predictive Control (MPC) approach adopted here is specifically tailored for centralized scheduling applications, where computational efficiency and deterministic optimization are prioritized. In contrast, distributed scheduling systems entail higher-dimensional optimization problems, leading to substantially increased computational complexity and coordination challenges. To address these limitations, future research should investigate enhanced MPC algorithms with decentralized capabilities or hybrid methodologies that integrate Deep Reinforcement Learning (DRL) for improved scalability and adaptability. The key contribution of this work lies in its hierarchical optimization framework for intra-day scheduling, coupled with seamless integration into real-time optimization strategies, thereby enhancing operational efficiency and decision-making precision.

4. How accurate is the proposed forecasting model? A comparative analysis with other existing forecasting methods is suggested to highlight its effectiveness.

Response: Thanks for your good suggestion. We evaluate commonly employed forecasting models, analyzing their respective application scenarios and distinguishing characteristics, as summarized in Table 3. Through this comparative analysis, Model Predictive Control (MPC) emerges as the optimal approach for our research framework. We added Table 3 and the following sentence to paragraph 1 of the 4.1 section:

5. It is recommended to include a brief discussion on future work in the conclusion to provide a forward-looking perspective.

Response: Thanks for your good suggestion. We have modified the manuscript and added the following sentence to the 6-th section:

Reviewer #2:

Thank you for taking time out of your busy schedule to read our research work again. Your valuable suggestions are very helpful for the revision of our paper. We have read your opinions carefully and revised our manuscript. We hope this revision can meet your standards. Here are the point-to-point responses.

1. The CNN-ATT-BiLSTM model proposed in the paper is only compared with CNN and BiLSTM, and other mainstream prediction models (such as Transformer, GRU, etc.) or hybrid models (such as CNN-LSTM) are not included as baselines, making it difficult to comprehensively demonstrate its superiority.

Response: We are very grateful for your suggestion. We have increased the comparison with the hybrid model CNN-LSTM and modify Figure 3-4 and Table 2 in 3.3 section. We also modified the manuscript and added the following sentence to last paragraph of 3.3 section:

2. The data only mentions "Data will be made available on request", but the specific sources, time range, or preprocessing methods are not clearly stated.

Response: Thanks for your good suggestion. We have modified the manuscript and added the following sentence to paragraph 1 of the 5-th section:

3. The scenario comparison only shows the results of the total cost and carbon emissions, lacking a quantitative analysis of the specific scheduling strategies (such as the start-stop timing of equipment and the charging and discharging modes of energy storage). It is recommended to supplement dynamic diagrams of the scheduling process during typical periods (such as the peak and valley electricity price periods) and explain how DR and the carbon trading mechanism collaborate to optimize the output of equipment.

Response: We are very grateful for your suggestion. We increase the peak and valley load scheduling under demand response. We also added Fig 6, Fig 7 and following sentence to 4.2 section:

4. The literature review part does not fully cover relevant studies in recent years (such as multi-time-scale scheduling based on reinforcement learning, integration of hydrogen energy storage, etc.), and there is a lack of critical analysis of the disadvantages of existing methods. It is recommended to expand the scope of the literature and clarify the unique contributions of the model in this paper compared with similar works.

Response: We are very grateful for your suggestion. We have modified the manuscript and added relevant literature.

5. The paper assumes that the response rates of all equipment are fixed (such as the ramp rate of the Micro-Turbine (MT) and the charging and discharging efficiency of energy storage), but does not discuss parameter sensitivity or conduct robustness tests (such as the system stability when the prediction errors of wind and solar power increase).

Response: We appreciate this insightful observation. The current study indeed adopts fixed efficiency and ramp rates for modeling simplicity, as our primary focus was to establish a baseline framework for intra-day scheduling, coupled with seamless integration into real-time optimization strategies, thereby enhancing operational efficiency and decision-making precision. We introduced slack variables to ensure feasible dispatch across stages and added the following sentence to last paragraph of 4.1 section:

We fully acknowledge that parameter sensitivity and prediction error robustness warrant further investigation. These factors will be prioritized in our future work, particularly for real-time control scenarios where dynamic responses are critical. We have modified the manuscript and added the following sentence to the 6-th section:

6. It is necessary to unify the format of the charts, supplement the necessary annotations, and check the continuity of the formula numbering.

Response: Thanks for your good suggestion. We added some charts, reprocessed the previously unclear graphs, and rechecked the formula numbering. After a comprehensive inspection, we confirm that all the specified elements meet the required standards and are correct. After comprehensive examination, we confirm that all specified elements meet the required standards and are correct as presented.

Once again, thank you for taking the time to review our work. Your constructive comments and suggestions would help us to develop research work and improve the quality of the paper in depth.

Best regards

Yours sincerely

Dengfeng Zhang

Institute of Intelligent Manufacturing

Nanjing Tech University

Jiangsu Province, China, 211816

Email: zhdfnjtech@163.com

---

## [Decision Letter · Decision Letter 1]

A Multi-Time Scale Rolling Optimization Framework for Low-Carbon Operation of CCHP Microgrids with Demand Response Integration

PONE-D-25-20221R1

Dear Dr. Zhang,

We’re pleased to inform you that your manuscript has been judged scientifically suitable for publication and will be formally accepted for publication once it meets all outstanding technical requirements.

Kind regards,

Zhengmao Li

Academic Editor

PLOS ONE

Additional Editor Comments (optional):

Reviewers' comments:

Reviewer's Responses to Questions

**Comments to the Author**

Reviewer #1: All comments have been addressed

Reviewer #2: (No Response)

2. Is the manuscript technically sound, and do the data support the conclusions?

Reviewer #1: Yes

Reviewer #2: (No Response)

3. Has the statistical analysis been performed appropriately and rigorously?

Reviewer #1: Yes

Reviewer #2: (No Response)

4. Have the authors made all data underlying the findings in their manuscript fully available?

Reviewer #1: Yes

Reviewer #2: (No Response)

5. Is the manuscript presented in an intelligible fashion and written in standard English?

Reviewer #1: Yes

Reviewer #2: (No Response)

Reviewer #1: (No Response)

Reviewer #2: (No Response)

**Do you want your identity to be public for this peer review?** For information about this choice, including consent withdrawal, please see our Privacy Policy

Reviewer #1: No

Reviewer #2: No

---

## [Editor Report · Acceptance letter]

PONE-D-25-20221R1

PLOS ONE

Dear Dr. Zhang,

I'm pleased to inform you that your manuscript has been deemed suitable for publication in PLOS ONE. Congratulations! Your manuscript is now being handed over to our production team.

Kind regards,

on behalf of

Dr Zhengmao Li

Academic Editor

PLOS ONE